# Integrative analyses reveal signaling pathways underlying familial breast cancer susceptibility

Stephen R Piccolo[1,2,3], Laura M Hoffman[4,5], Thomas Conner[4], Gajendra Shrestha[1], Adam L Cohen[4,6], Jeffrey R Marks[7], Leigh A Neumayer[8], Cori A Agarwal[8], Mary C Beckerle[4,5,9], Irene L Andrulis[10], Avrum E Spira[2], Philip J Moos[1], Saundra S Buys[4,6], William Evan Johnson[2,9] & Andrea H Bild[1,9,*]

## Abstract

The signaling events that drive familial breast cancer (FBC) risk remain poorly understood. While the majority of genomic studies have focused on genetic risk variants, known risk variants account for at most 30% of FBC cases. Considering that multiple genes may influence FBC risk, we hypothesized that a pathway-based strategy examining different data types from multiple tissues could elucidate the biological basis for FBC. In this study, we performed integrated analyses of gene expression and exome-sequencing data from peripheral blood mononuclear cells and showed that cell adhesion pathways are significantly and consistently dysregulated in women who develop FBC. The dysregulation of cell adhesion pathways in high-risk women was also identified by pathway-based profiling applied to normal breast tissue data from two independent cohorts. The results of our genomic analyses were validated in normal primary mammary epithelial cells from high-risk and control women, using cell-based functional assays, drug-response assays, fluorescence microscopy, and Western blotting assays. Both genomic and cell-based experiments indicate that cell–cell and cell–extracellular matrix adhesion processes seem to be disrupted in non-malignant cells of women at high risk for FBC and suggest a potential role for these processes in FBC development.

**Keywords** breast cancer; cellular adhesion; disease susceptibility; multiomic analysis, signaling pathways
**Subject Categories** Genome-Scale & Integrative Biology; Cancer
**Mol Syst Biol. (2016) 12: 860**

## Introduction

The biological basis for breast cancer has been clarified by studies that have described risk factors that influence tumor development. Such risk factors include genetic risk alleles and expression levels for individual genes (Miki *et al*, 1994; Wooster *et al*, 1994). However, breast tumorigenesis arises from a complex interplay among genes within signaling pathways. When one or more components of a given network are disrupted, cells may have a greater propensity to transform, proliferate, and invade surrounding tissue (Hanahan & Weinberg, 2011). Importantly, dysregulation of different genes within the same pathway may have a similar impact on downstream pathway function (Yarden & Sliwkowski, 2001; Wood *et al*, 2007). Accordingly, to gain a broader perspective of the molecular aberrations that contribute to tumor development, it is valuable to evaluate data at the pathway level rather than evaluating single genes. Furthermore, by evaluating pathway activity using multiple types of "omic" data—including both DNA variation and gene expression changes—it is possible to develop a more complete picture of disease mechanisms. In addition, by examining pathway dysregulation in normal cells, one can better understand the processes by which germline variation leads to tumorigenesis; such effects should be observable not only in normal mammary epithelial cells but also in peripheral blood cells, which can be obtained less invasively (Liew *et al*, 2006; Mohr & Liew, 2007).

We hypothesized that by comparing molecular profiles for women from high-risk families who have developed breast cancer against women from high-risk families who have not developed breast cancer, we might identify biological pathways that play a role in development of familial breast cancer (FBC). In addition, we hypothesized that transcriptomic profiling—summarized at the pathway level—would enable us to account for common downstream effects of different germline variants across many women

1 Department of Pharmacology and Toxicology, University of Utah, Salt Lake City, UT, USA
2 Division of Computational Biomedicine, Boston University School of Medicine, Boston, MA, USA
3 Department of Biology, Brigham Young University, Provo, UT, USA
4 Huntsman Cancer Institute, Salt Lake City, UT, USA
5 Department of Biology, University of Utah, Salt Lake City, UT, USA
6 Department of Medicine, University of Utah, Salt Lake City, UT, USA
7 Department of Surgery, Duke University School of Medicine, Durham, NC, USA
8 Department of Surgery, University of Utah, Salt Lake City, UT, USA
9 Department of Oncological Sciences, University of Utah, Salt Lake City, UT, USA
10 Lunenfeld-Tanenbaum Research Institute, Mount Sinai Hospital, Toronto, ON, Canada
*Corresponding author. Tel: +1 801 581 6353; Fax: +1 801 581 5111; E-mail: andreab@genetics.utah.edu

who develop FBC. In other words, even though individuals who develop FBC may differ in the specific germline variants that they carry, those variants may affect similar signaling networks and have similar downstream transcriptional consequences.

Transcriptional profiling has been used widely to improve our understanding of the molecular basis of cancer phenotypes (Miller et al, 2005; Bild et al, 2006; Huang et al, 2007; Langenau et al, 2007; Rhodes et al, 2007; Liu et al, 2008; Waddell et al, 2008; Wong et al, 2008; Barbie et al, 2009; Ooi et al, 2009; Singh et al, 2009; Zhang et al, 2009). In normal cells, gene expression levels reflect functional germline genetic (and epigenomic) variation and thus may be useful as a surrogate for germline variation. Associations between genetic and transcriptomic variation were initially mapped in yeast where it was shown that genetic variants in parental yeast influence expression traits in progeny (Steinmetz et al, 2002). Subsequent studies have identified thousands of expression quantitative trait loci in the human genome (Morley et al, 2004; Schadt et al, 2008), and additional research has demonstrated that global mRNA expression patterns can reflect heritable disease susceptibility (Mohr & Liew, 2007; Cookson et al, 2009). For example, in individuals who carry BRCA1/2 mutations, expression patterns in lymphoblastoid cells are distinct from those in individuals who do not carry these mutations (Waddell et al, 2008). Such downstream effects have also been observed in tumors (Hedenfalk et al, 2001), suggesting that gene expression levels can reflect germline variation.

We performed a gene expression analysis to profile peripheral blood mononuclear cells (PBMCs) in two patient cohorts that included women who had a family history of breast cancer; approximately half of these women had developed breast cancer, while the others had not. We also examined germline DNA variation in a subset of these samples to test for consistency of pathway dysregulation across these genomic data types. To study the biological mechanisms of FBC, we mapped the data to 932 canonical signaling pathways (Subramanian et al, 2005; Kanehisa et al, 2006; Taube et al, 2010; Byers et al, 2013) and used these data to discover pathway-specific patterns that differed between individuals who developed FBC and those who did not. Multiple pathways related to cell–cell and cell–extracellular matrix (ECM) adhesion showed significant differences in gene expression and germline variation. Next, to verify whether these observations generalize to breast tissue, we performed a pathway analysis on two gene expression data sets that profiled normal breast tissue from women who had a family history of breast cancer and/or carried a mutation in BRCA1/2 and compared them against control women from the same studies. Again, various cell adhesion pathways attained statistical significance in these data sets. Based on these findings, we used cell-based functional assays and fluorescence microscopy to compare cell adhesion properties of normal mammary epithelial cells between women who had undergone prophylactic mastectomy due to a breast cancer family history and women who had undergone a breast reduction surgery for reasons unrelated to breast cancer risk (these served as controls).

In support of our genomic findings, we observed significant differences in cell–cell and cell–ECM phenotypes for the prophylactic mastectomy cells compared with the controls. Specifically, normal cells from FBC women showed: (i) decreased cell adhesion abilities, (ii) increased cell death when treated with drugs that modulate cell

adhesion capacity, and (iii) aberrant morphological features. Taken together, these findings, derived from multiple computational and experimental analyses on multiple tissue types, implicate dysregulated cell adhesion pathways in FBC development.

# Results

## Overview of approach to identify familial breast cancer susceptibility pathways by multi-omic profiling

Our initial analyses aimed to identify pathways that differed between women who developed FBC and those who did not, across multiple types of omic data. Because germline variation drives transcriptional changes that should be observable in both peripheral blood cells and breast cells, we obtained gene expression data for PBMCs and normal breast cells across four independent cohorts (Fig 1B and C). We also obtained exome-sequencing data for a subset of the PBMC samples to identify pathway-level germline DNA variation associated with FBC development. We analyzed the omic data using pathway-based approaches, identifying pathways that consistently showed dysregulation between women who were or were not affected by FBC (Fig 1A). In this sense, our genomic data enabled us to generate and filter hypotheses regarding biological processes that play a role in FBC development; we then validated these findings in the laboratory using cell-based assays (Fig 1C).

## Pathway-level evaluations of peripheral blood cells for women from breast cancer families

For patients from Utah (USA) and Ontario (Canada), we used gene expression profiles of peripheral blood mononuclear cells to identify signaling pathways differentially expressed between women who developed FBC and women who had a family history of breast cancer but who did not develop cancer by at least age 55. We also sequenced the exomes of 35 Utah patients and assessed whether pathway-level germline variation differed between these patient groups. For the expression data, we mapped gene-level values to 932 curated signaling pathways and used the Support Vector Machines algorithm (Vapnik, 1998) to test how well a multigene classifier —based only upon genes from a given pathway—could differentiate between women who developed FBC and women who did not. Pathways were ranked according to their accuracy at distinguishing FBC women versus controls (Fig 1A). For mutation data, we compared the number of FBC samples that contained a variant in a given pathway against the number of control samples with a pathway variant (see Materials and Methods).

Using the Utah data as a training set and the Ontario data as a test set, we focused on the 45 pathways that consistently and significantly discriminated between affected FBC women and controls for both cohorts and for both types of omic data (rank P-value < 0.05; Dataset EV1). Because many canonical pathways reflect similar biological functions, we manually categorized the pathways according to biological themes that describe their activity (Appendix Table S2). As shown in Fig 2A, the most significant cancer-related biological themes were cell adhesion, MAPK signaling, and cell cycle regulation (Hanahan & Weinberg, 2011). Figure 2B and C shows gene expression and DNA mutation

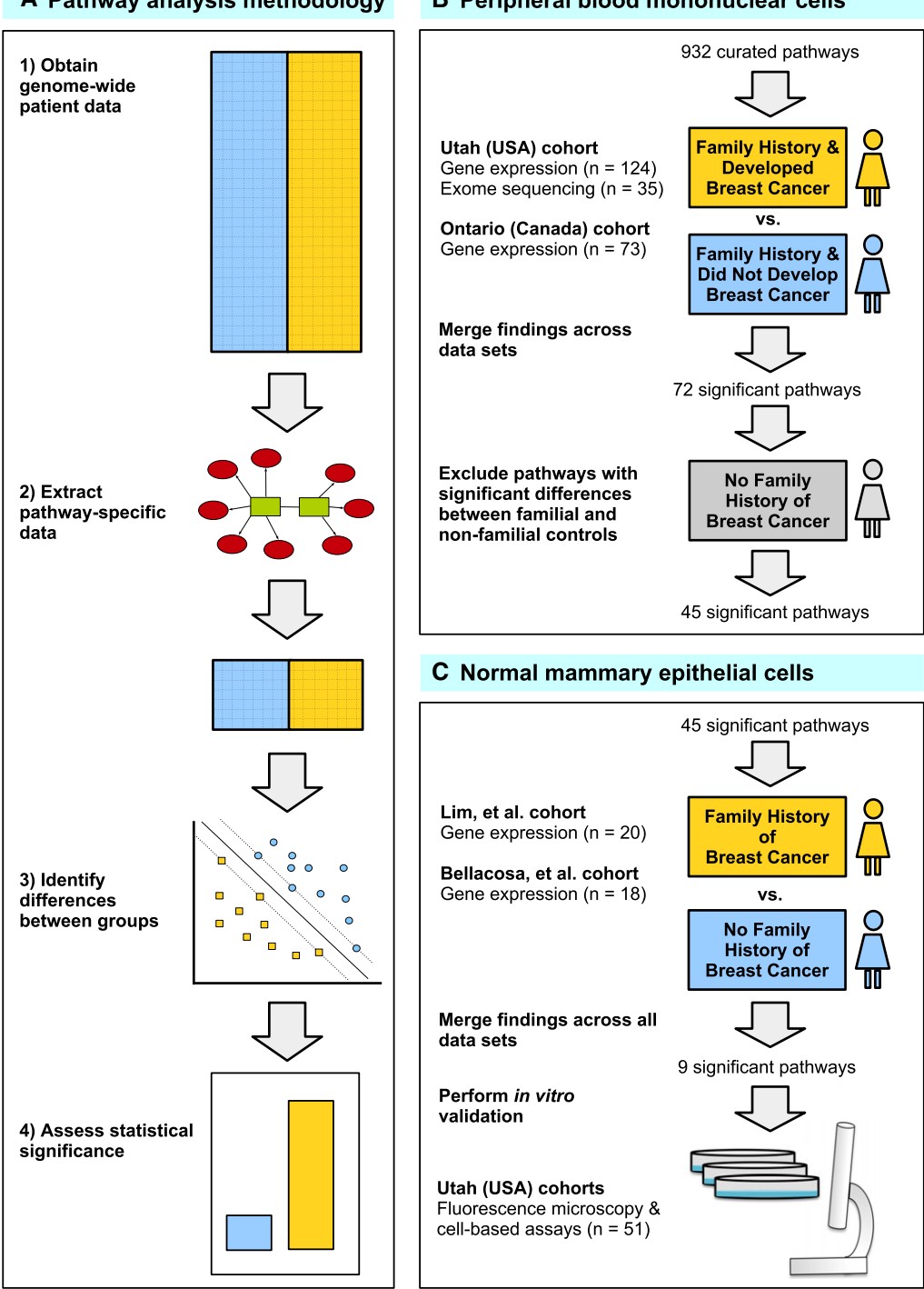

**Figure 1.   Flow chart illustrating the experimental design of this study**.

A   We used pathway-based analytic approaches to identify biological processes that may be disrupted in women who develop familial breast cancer (FBC). Having collected genomewide data, we filtered the data to include only genes associated with a given pathway. For each pathway, we identified differences between individuals who developed FBC and those who did not, using either the Support Vector Machines algorithm (gene expression data) or Barnard's exact test (DNA variant data). We considered the most statistically significant pathways to be candidates for further investigation.

B   We profiled peripheral blood mononuclear cells using gene expression microarrays and exome sequencing and identified pathways that were consistently significant across these data sets. To reduce the chance that our findings were influenced by treatment effects, we excluded pathways that showed significant differences between familial and non-familial controls.

C   For the remaining pathways, we identified those that showed significant differences in two gene expression data sets representing primary mammary epithelial cells. To validate these findings, we used cell-based assays and fluorescence microscopy to profile an additional collection of normal breast cells.

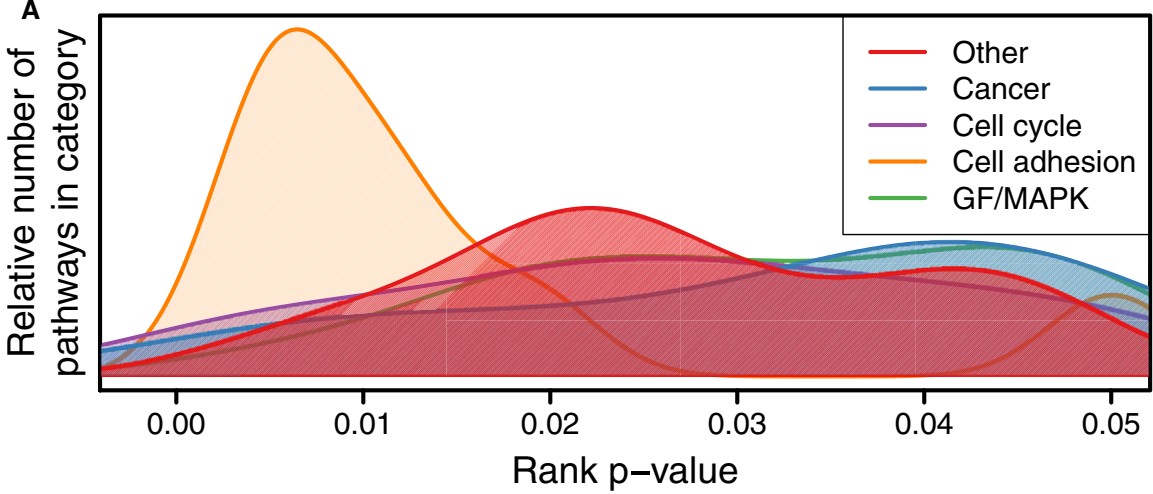

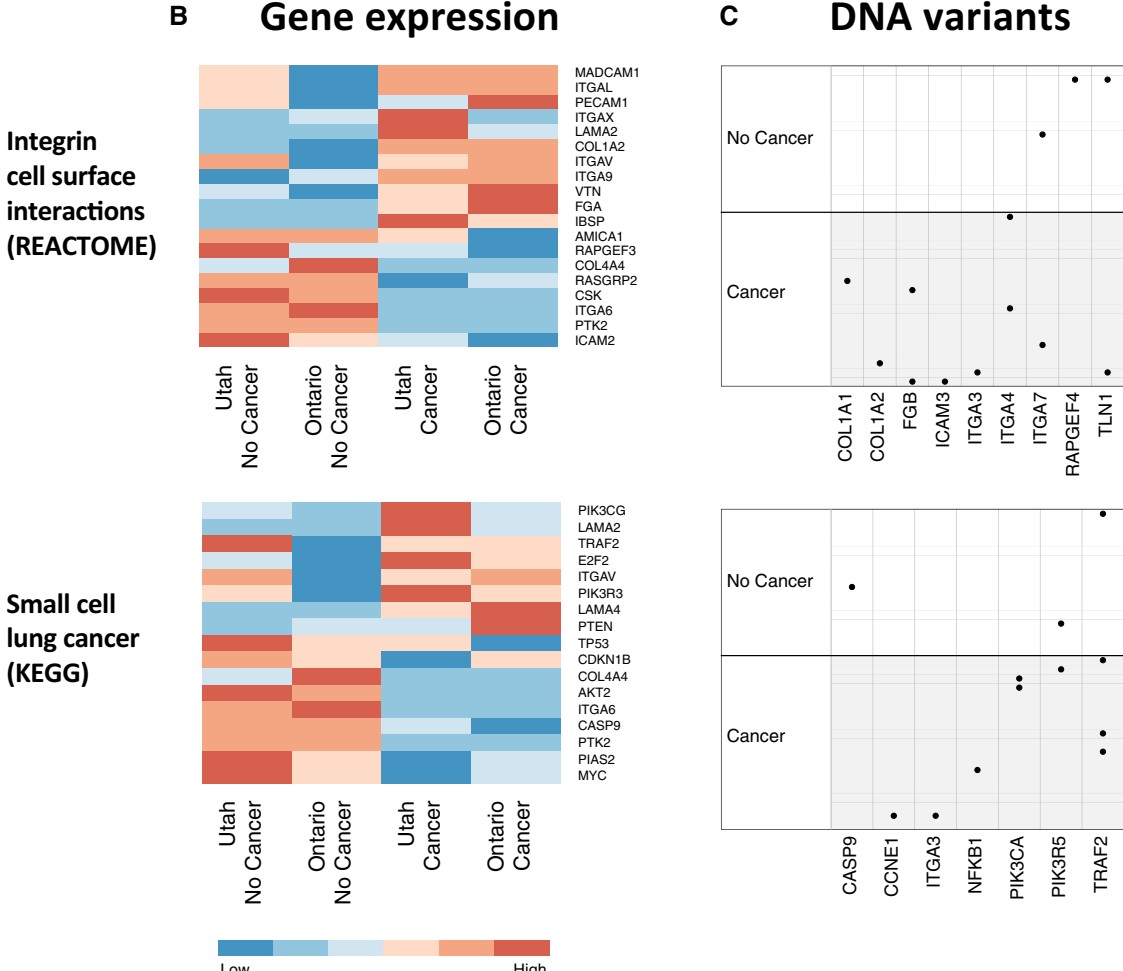

**Figure 2.  Overview of top pathways for which gene expression levels and mutation status differed significantly between FBC women and controls in the Utah and Ontario cohorts.**

A   Biological processes associated with pathways that attained a rank *P*-value < 0.05.

B   Heatmaps show median expression levels for Utah and Ontario women, respectively, who developed FBC and for women who did not. Only genes that exhibited a consistent fold change across the cohorts are shown.

C   Per-sample DNA variants observed in these pathways are shown. Black dots indicate samples that carried a likely pathogenic in the genes that are shown. Only genes for which at least one variant was observed are shown.

patterns for the two pathways that performed best overall in these analyses (Datasets EV2–EV4). The REACTOME *Integrin Cell Surface Interactions* pathway ($P = 0.004$) characterizes interactions that occur between ECM macromolecules and cell surface proteins to provide a substrate for epithelial cells (Faull & Ginsberg, 1996). The KEGG *Small Cell Lung Cancer* pathway ($P = 0.005$) characterizes various biological processes that can influence tumorigenesis—including cell–ECM interactions, cell proliferation, and cell cycle regulation. Various other pathways related to cell adhesion—including KEGG *Focal Adhesion* ($P = 0.012$) and REACTOME *Cell Surface Interactions at the Vascular Wall* ($P = 0.007$)—also performed well in our analyses. Together, these findings suggest that perturbation of normal cell adhesion activity may lead to an increased risk of breast tumors in FBC women.

To assess the robustness of these results, we repeated the pathway-filtering steps using three alternative methods for combining *P*-values across data sets (see Materials and Methods) and then corrected for multiple tests using Storey's *q*-value method (Storey, 2003). Each of these approaches resulted in a larger number of candidate pathways, and the same pathways that we had identified using the rank-based approach were among the top pathways identified with these approaches (Datasets EV5–EV7). Lastly, to further interrogate whether cell adhesion pathways were significant in specific patient subsets independently, we repeated the analyses but separated the data according to whether each patient carried a BRCA1/2 mutation or did not (BRCAX) (Appendix Figs S4 and S5). Comparing cancer vs. no cancer samples within each of these groups, we found that a variety of cell adhesion-related pathways attained statistical significance (Datasets EV8 and EV9). Accordingly, although samples sizes are relatively small for these subgroup comparisons, these results suggest that cell adhesion processes may play a role in breast cancer risk for both groups. Because our overarching goal was to identify pathways that are common to and not specific to either BRCA1/2 or BRCAX carriers alone, we excluded any pathway that showed significant ($P < 0.05$) differences between BRCA1/2 and BRCAX individuals (Datasets EV10 and EV11) from further analysis.

### Evaluation of familial breast cancer susceptibility pathways via profiling of normal breast tissue

We next applied our pathway analysis approach to two additional gene expression data sets to evaluate which pathways showed significant dysregulation in *normal* breast tissue (Lim *et al*, 2009; Bellacosa *et al*, 2010). Dataset EV12 lists pathways that showed an ability to discriminate between (i) women who had a family history of breast cancer and/or carried a BRCA1/2 mutation and (ii) controls in these data sets. Of the 45 pathways identified in the previous analysis, 9 showed significant differences for these two data sets as well as when combining evidence across all 5 data sets (*P*-value < 0.05). Again, the highest ranking pathways were REACTOME *Integrin Cell Surface Interactions* ($P = 0.038$ for Lim *et al* and $P = 0.030$ for Bellacosa *et al*) and KEGG *Small Cell Lung Cancer* ($P = 0.007$ and 0.003, respectively) (Fig 3; Datasets EV13–EV16). Thus, expression patterns in these pathways for both peripheral blood and normal breast cells were remarkably consistent, again highlighting the potential role of cell adhesion signaling in FBC susceptibility. Although pathway perturbation mechanisms

may vary from one individual to the next, our results consistently point to disrupted cell adhesion mechanisms as an indicator of breast cancer risk.

To further evaluate the relationship between gene expression and protein levels, we performed a Western blotting analysis using snap-frozen tissue from an independent cohort of breast epithelial tissues that consisted of women undergoing prophylactic surgeries for BRCA1/2 mutation and/or high-risk status and for controls who underwent breast reduction surgeries for non-cancer-related reasons ($n = 27$). Similar to the gene expression data, vitronectin (VTN) showed a significant increase in protein levels, while F-actin showed a significant decrease for FBC women compared to controls (*P*-value < 0.05), indicating that gene and protein expression for a subset of genes highlighted in the genomic analyses described above are concordantly dysregulated. Furthermore, FAK/PTK2 and PTEN protein levels trended lower in FBC patient samples compared to controls (Fig EV1, Appendix Table S3).

### Aberrant cell morphology in normal mammary epithelial cells of women from FBC families

As our gene expression and DNA mutation analyses consistently indicated cell adhesion pathways as a candidate mechanism contributing to FBC, we investigated whether differences in cell adhesion phenotypes would be observable in normal breast tissue. Previous studies have associated cell adhesion perturbations and dysregulation of ECM components with breast tumorigenesis (Schor *et al*, 1985; Weaver *et al*, 1996, 1997, 2002; Schor & Schor, 2001; Wang *et al*, 2002; Paszek *et al*, 2005; Kass *et al*, 2007; Levental *et al*, 2009; Tanner *et al*, 2012); however, these pathways have not previously been linked to breast cancer susceptibility. Accordingly, we used normal mammary epithelial cells—obtained after prophylactic mastectomies from women who had a family history of breast cancer—and compared them to cells from control women who underwent breast reduction surgeries. We tested whether cell–cell and cell–ECM phenotypes differed between the groups. Irrespective of surgery type, all patient samples were processed using similar procedures. Cold ischemia times were on average < 30 min, and growth conditions for these viable tissues were the same for both surgical procedures.

First, we asked whether we could detect qualitative differences in cytoskeletal cell–cell adhesion phenotypes. Normal primary mammary epithelial cells from 24 patient cultures were seeded onto glass slides, grown for 3–5 days, then fixed, stained, and imaged using fluorescence microscopy. Figure 4A shows cells stained for F-actin, focal adhesions, and nuclei from ten of the patients, which had been assessed in a blinded manner for having distinctive cell phenotypes. Cell morphology ranged from compact clusters of tightly bunched cells to dispersed single cells with well-developed actin cytoskeletons and adhesion sites. After unblinding, we found that the control cell cultures more frequently clustered together and displayed actin filaments and focal adhesions that resemble typical observations for normal mammary epithelial cell cultures; however, many of the FBC cultures exhibited larger, well-spread cells with more visible actin cytoskeletons and fewer cell–cell contacts. We quantified these observations (see Materials and Methods) for all microscope fields ($n = {\sim}10$) from the 10 patients identified above and observed significant differences ($P < 0.05$) between the groups in estimated cell size, F-actin staining, and focal adhesion staining (Fig 4B).

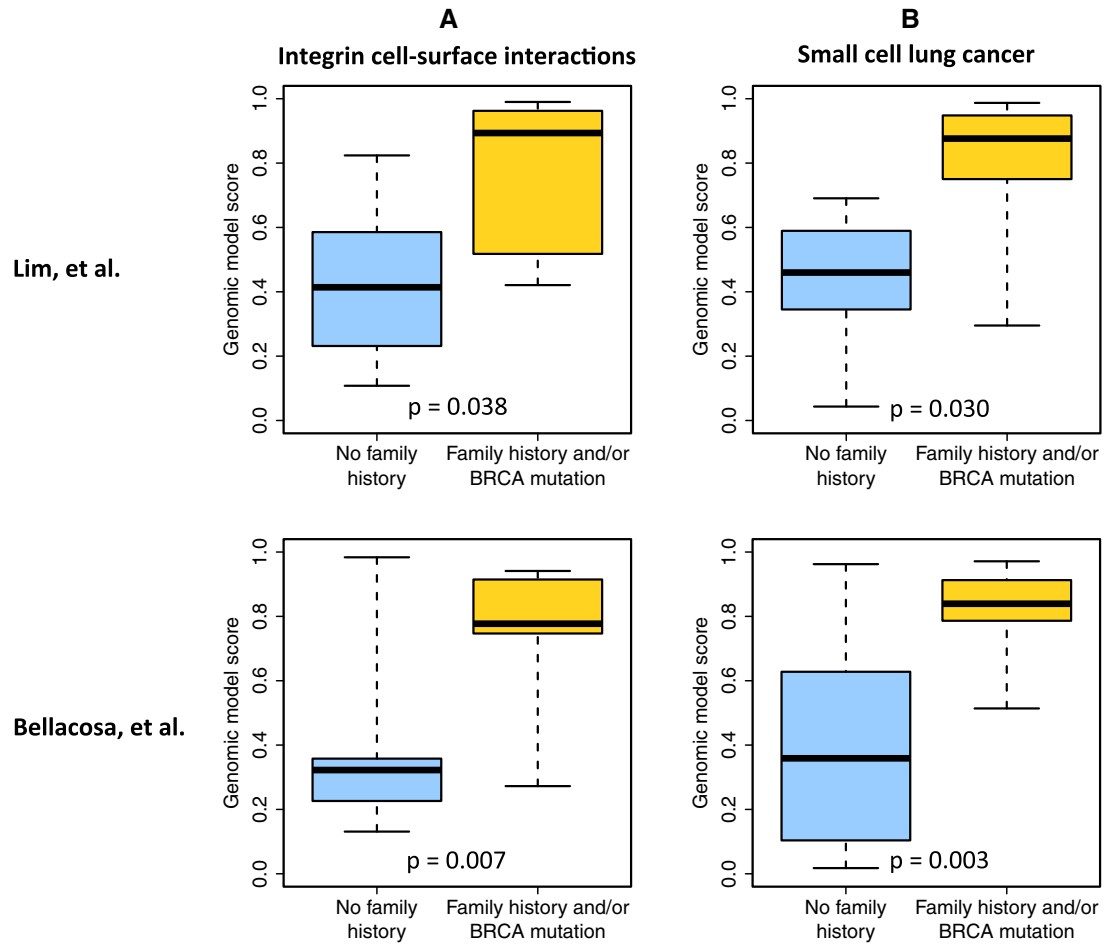

**Figure 3.  Summary of pathway-level results that included non-FBC controls and normal breast gene expression data.**

A   Cross-validated estimates that each patient from the Lim *et al* and Bellacosa *et al* cohorts had a family history of breast cancer and/or carried a BRCA1/2 mutation. These estimates were derived from genes in the REACTOME *Integrin Cell Surface Interactions* pathway.

B   Estimates for the same patients using genes from the KEGG *Small cell lung cancer* pathway.

Data information: The boxes represent the the interquartile range of the "Genomic model score" values. The whiskers extend the the most extreme data points.

## Aberrant cell adhesion properties in normal mammary epithelial cells of women from FBC families

To further evaluate the functional role of cell adhesion in FBC, we used an *in vitro* assay to assess the cells' ability to adhere. We allowed the mammary epithelial cell cultures to adhere to laminin-coated plates for three hours to test for cell–ECM interaction and adherence. We then quantified the number of cells that adhered to the plates and observed a modest but significant decrease in adherent cells for FBC samples compared to controls (Fig 5A, *P*-value = 0.02), again supporting the findings from our genomic studies. Of note, the adhesion phenotype observed within the primary cultures may be time-in-culture dependent, as we did not see a difference in focal adhesions for FBC cells grown longer term in culture, which is consistent with our fluorescence microscopy observations. This subtle variance suggests that differences in cell–ECM adherence may be observable in short-term culture acute settings but may be compensated for, and lost, in longer term culture experiments.

## Pharmacological evidence for aberrant cell adhesion properties in normal breast epithelial cells of FBC women

We treated breast epithelial cells with PF573228, a focal adhesion kinase (FAK) inhibitor (Slack-Davis *et al*, 2007), as FAK (PTK2) is critical in focal adhesions that form among cells attaching to the extracellular matrix, as well as in cell migration. If normal epithelial cells from FBC women are "primed" with decreased cell adhesion, then inhibiting FAK in these cells would potentially have increased potency when compared to control cells. We found that cells from FBC patients were significantly more sensitive than controls to this drug (Fig 5B), suggesting that there are deficits in adhesion properties in FBC patient cells and indicating a role of FAK-related signaling and the actin cytoskeleton in FBC development.

As a control experiment, we assessed cell proliferation when cells from the same cultures were treated with gefitinib and afatinib, which target growth factor-related pathways but are not closely related to cell adhesion, but we saw no difference in response for these growth control drugs (Fig EV2).

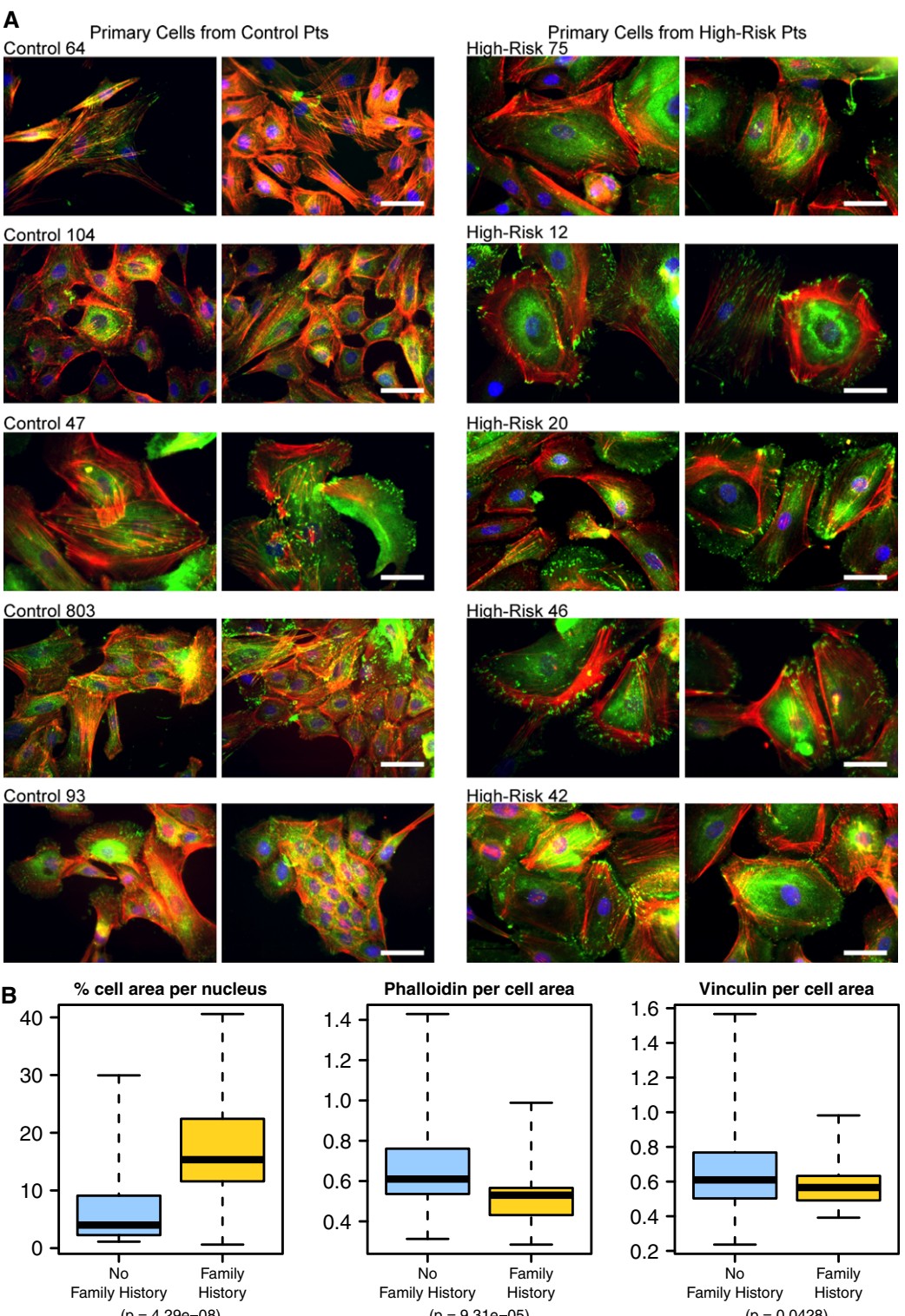

**Figure 4. Fluorescence microscopy images of primary breast epithelial cultures for FBC women and controls.**

A   Primary (non-malignant) mammary epithelial cells from breast reduction patients with no known family history of breast cancer and from prophylactic mastectomy patients who had a breast cancer family history ("high risk") were cultured on glass slides for 3–5 days and subsequently fixed and stained for F-actin (red, phalloidin), focal adhesions (green, vinculin) and nuclei (blue, Dapi). Shown are five cell populations (two fields each) from each group that had been identified in a blinded manner as having distinctive cell phenotypes. Scale bar = 50 μm.

B   Box plots showing the results of quantitative comparisons for all microscope fields ($n$ = ~10) from each of the samples shown. Samples from high-risk patients were more spread apart and expressed higher levels of F-actin. See Materials and Methods for details about how quantitative metrics were derived. The boxes represent the the interquartile range of the respective values. The whiskers extend the the most extreme data points.

## A Cell adhesion assay

## B FAK inhibitor assay

**Figure 5.  Cell-based assays show aberrant cell adhesion in normal breast epithelial cells for women with a high risk of breast cancer.**

A  A cell adhesion assay was used to compare extracellular matrix adhesiveness in normal, primary breast cells in women who did or did not have a family history of breast cancer. Cells from women who had a family history of breast cancer were significantly less adherent than cells from women who did not have a family history of breast cancer.

B  A drug-response assay was used to evaluate responsiveness to PF573228, a FAK inhibitor in μm concentrations. Normal, breast epithelial cells were obtained from high-risk women who had undergone prophylactic surgery and compared against cells from women who had undergone non-risk-related, breast reduction surgery. The prophylactic samples were significantly more sensitive to PF573228 than breast reduction samples. Response values indicate the drug concentration that induces a response that reaches half of its maximal effect.

Data information: The boxes represent the the interquartile range of the respective values. The whiskers extend the the most extreme data points.

In sum, our *in silico* and *in vitro* observations suggest that alterations to cell adhesion regulatory pathways may lead to distinct cell phenotypes in women with a family history of breast cancer, that these alterations may lead to decreased cell–cell contact disposition in response to growth and that this functional mechanism may play a role in FBC development.

## Discussion

Since the discovery of *BRCA1* and *BRCA2* as breast cancer susceptibility genes (Miki *et al*, 1994; Wooster *et al*, 1994), much focus has been placed on discovering additional DNA variants that are associated with FBC development. Ongoing efforts to genotype ever-larger cohorts have also yielded common susceptibility variants (Michailidou *et al*, 2013); however, in total, known susceptibility variants explain < 30% of familial risk (Stratton & Rahman, 2008). Accordingly, new complimentary approaches are needed to identify genomic factors that drive FBC risk. In our study, we took a multiomic approach and searched for multigene patterns associated with FBC development, irrespective of *BRCA1/2* mutation status. Our approach is based on the premise that germline genetic and epigenomic variations cause gene expression changes in normal cells that reflect a person's risk for eventual tumor development. Upon examining gene expression levels and protein-coding variants for women who did or did not develop FBC, we identified signaling pathways with consistent differences between the groups, including pathways related to cell adhesion, integrin signaling, and growth signaling. We also evaluated normal breast cells using fluorescence microscopy, functional assays, and pharmacologic assays; each provided additional evidence that cell adhesion pathways are dysregulated in high-risk women. These findings complement prior research, which has shown that blood-derived molecular signatures reflect dysregulated molecular processes in breast tissue (Sharma *et al*, 2005; Aarøe *et al*, 2010; Tudoran *et al*, 2014), that dysregulation of cell adhesion genes can be identified in blood cells from breast cancer patients (Tudoran *et al*, 2014), and that germline variants in cell adhesion pathway genes contribute to breast cancer risk (Ayala *et al*, 2003; Langsenlehner *et al*, 2006; Liu *et al*, 2013). Our analysis extends these observations by aggregating pathway-level genomic data from four independent patient populations (five data sets total) —including two that characterized expression within normal breast cells. Importantly, we also used our genomic findings to guide laboratory-based experiments that together provide a detailed characterization of how dysregulation of cell adhesion pathways in high-risk women may modulate cell–cell and cell–ECM properties, as well as differential responses to changes in these interactions. Such integrative approaches are imperative for deciphering mechanisms that influence disease risk.

Summarizing genomic data at the pathway level allows data to be placed in biological context and facilitates experimental follow-up. However, one challenge with pathway-based bioinformatics analyses is that many candidate pathways typically emerge. We addressed this issue by examining five genomic data sets to identify

themes that were common across them. Importantly, many of the pathways we identified reflect similar biological processes. Subsequent laboratory-based assays supported our genomic findings and further implicated cell adhesion pathways as being dysregulated in high-risk patient tissues. These findings demonstrate the value of leveraging multiple pathways, cohorts, tissue types, and data types in studies with small or moderate sample size to identify functional dysregulation associated with a given phenotype—in this case, familial breast cancer.

Although we focused primarily on cell adhesion as a candidate susceptibility mechanism, additional biological processes performed consistently well across our genomic analyses—in particular, pathways that are known to play diverse roles in tumor development (Fig 2A and Appendix Table S2). These pathways regulate processes such as cellular proliferation, differentiation, and the cell cycle and include well-known cancer genes such as *ERBB2*, *EGFR*, *PIK3CA*, and *JUN*. These observations also are consistent with a recent prospective study that showed enrichment of cancer (and cell adhesion) pathways in blood-derived DNA methylation profiles for women who developed familial breast cancer (Xu *et al*, 2013). Together, our findings suggest that aberrant signaling within cancer pathways in peripheral blood may be a sign of eventual tumor development in general.

In this study, we focused on women who had at least two-first-degree relatives who had been diagnosed with cancer. Such women have a considerably higher risk of developing breast cancer than women who have only a single affected first-degree relative. However, our findings may also have relevance for the latter group. Future studies will be valuable in assessing whether our findings are relevant mainly to women who have a relatively strong family history of breast cancer or to a broader population.

Overall, our results suggest that cell adhesion-related pathways may exhibit different behavior between individuals who develop FBC and those who do not. Our evaluations of gene expression levels and germline variants across multiple cohorts identify an association between aberrant activity within these pathways and breast cancer susceptibility. The use of multi-omic data linked to functional and biological studies complements conventional approaches, such as genomewide association studies, to aid in identifying signaling networks that influence disease development. These findings may lead to improved methods of predicting the development of familial breast cancer, which could have significant implications for risk management.

# Materials and Methods

**Patient cohorts and ethics approval**

As summarized in Tables 1 and 2 and Appendix Table S1, we acquired blood samples for a cohort of 124 women from Utah, USA. Additionally, we recruited an independent cohort of 73 women from Ontario, Canada. Our objective was to determine whether pathway dysregulation in non-malignant cells correlates with the development of breast cancer in individuals who had a family history of breast cancer, with or without a BRCA1/2 mutation. Individuals who had a family history of breast cancer but had not developed cancer, or who did not have a family history but either had, or had

**Table 1. Overview of patients from Utah (USA) who provided a PBMC sample.**

| Family history of breast cancer | BRCA1/2 mutation | Developed tumor | # Patients | Median age |
|---|---|---|---|---|
| Yes | Yes | Yes | 16 | 59 |
| Yes | No | Yes | 23 | 59 |
| Yes | Yes | No | 18 | 60 |
| Yes | No | No | 26 | 63 |
| No | No | Yes | 22 | 66 |
| No | No | No | 19 | 58 |

The median age indicates the age in years at which blood was drawn.

**Table 2. Overview of patients from Ontario (CA) who provided a PBMC sample.**

| Family history of breast cancer | BRCA1/2 mutation | Developed tumor | # Patients |
|---|---|---|---|
| Yes | Yes | Yes | 11 |
| Yes | No | Yes | 17 |
| Yes | Yes | No | 14 |
| Yes | No | No | 18 |
| No | No | Yes | 8 |
| No | No | No | 5 |

not, developed cancer, served as controls. Accordingly, both the Utah and Ontario cohorts included women who (i) had a family history of breast cancer or did not, (ii) carried a pathogenic germline variant in BRCA1 or BRCA2 or did not, and (iii) had developed breast cancer or had not. Our definition of "family history of cancer" was limited to women who had at least two-first-degree relatives who had been diagnosed with cancer. Where possible, patients were matched according to age at which blood was drawn (see Table 1 and Appendix Fig S1). For individuals who had developed breast cancer, blood samples were collected at least 6 months after completing therapy.

Pathogenic mutation status in BRCA1/2 was identified via commercial, PCR-based genetic testing (Myriad Genetics). Technical specifications for these tests can be found here: https://www.myriad.com/lib/technical-specifications/BRACAnalysis-Technical-Specifications.pdf. Across our two cohorts, 30% of participants carried a known pathogenic mutation in BRCA1 or BRCA2. We labeled the remaining participants who had a family history of breast cancer but who did not carry a known pathogenic mutation in BRCA1 or BRCA2 as "BRCAX". Across the cohorts, 47% of participants had developed breast cancer, whereas the remaining participants had not developed breast cancer by at least 55 years of age; the median age among these women was 60 years. The samples showed no statistical difference in age at blood draw.

To control for confounding effects due to previous cancer treatments, our control group included 41 women from Utah and 13 women from Ontario who had no family history of breast cancer. Over half of these women had developed non-familial (sporadic) breast cancer (Tables 1 and 2). We included these cases to provide

confidence that gene expression differences were specific to FBC and driven primarily by germline susceptibility factors and not prior cancer treatment.

The Utah women were recruited via Huntsman Cancer Institute (High Risk Breast Cancer Clinic) under Institutional Review Board protocols (#00022886 and #00004965). We selected these women from among a larger pool of eligible participants who had a strong family history of breast cancer, based on whether we could obtain fresh blood cells at the time of recruitment. The Ontario women were recruited through the Breast Cancer Family Registry at Mt. Sinai Hospital. Informed consent was obtained from all participants.

### Gene expression data

In accordance with manufacturer (Becton-Dickinson) protocol, PBMCs were isolated from blood samples in cell preparation tubes. RNA was isolated using the RNAeasy Kit (Qiagen, Valencia, CA, USA) and hybridized to Affymetrix GeneChip Human Exon 1.0 ST microarrays.

We normalized the microarray data using the Single-channel Array Normalization method (Piccolo *et al*, 2012). Next, we excluded any microarray probe that overlapped with a known DNA variant or that was not classified as "green" in PLANdbAffy (Nurtdinov *et al*, 2010). We used a 10% trimmed mean to summarize the remaining probes at the gene level and excluded genes with four probes or fewer. We used ComBat (Johnson *et al*, 2007) to correct for any batch effects; because some Ontario samples were processed at two different facilities, we treated these samples as two separate batches.

We also excluded any gene whose expression might be confounded by immune activity or demographic/clinical variables associated with the patients. We applied a total lymphocyte enumeration test to whole blood cells for 22 samples from the Utah cohort to estimate the number of B cells, CD3-positive T cells, CD4-positive T cells, CD8-positive T cells, and NK cells. In addition, 63 patients from the Utah cohort responded to a health assessment survey that collected the following variables for each participant: age at diagnosis, age of first menstrual cycle, time since last menstrual cycle, age when menstruation ceased, level of education, religious preference, overall health status, marital status, level of physical activity, use of contraceptives, total number of pregnancies, first live birth age, last live birth age, number of live births, breastfeeding status, use of chemopreventive/hypertension/ anti-inflammatory drugs, tobacco use, alcohol use, occupational history, and history of immunological disorders. We used a multifactor ANCOVA test to identify genes ($n = 334$) for which gene expression levels correlated strongly with any of these factors ($P < 0.01$).

We used normal breast tissue expression data from Lim *et al* (Lim *et al*, 2009) (GSE17072) and Bellacosa *et al* (2010) (GSE19383). Using data preprocessed by the original authors, we compared gene expression levels between women who had a family history of breast cancer and/or who carried a pathogenic mutation in BRCA1/2 and control patients who did not meet these criteria.

### Exome-sequencing data

We used exome-capture DNA sequencing to profile peripheral blood cells from 35 of the Utah participants. Genomic DNA was hybridized

using *Agilent SureSelect Human All Exon v4 + UTRs* kits. Captured libraries were sequenced on an Illumina Hi-Seq 2000 instrument, and bar coding was used for multiplexing (seven lanes, five samples per lane). This process resulted in 101-bp paired-end reads (58,032,900 unique reads per sample).

We aligned raw sequencing reads to the *hg19* reference genome using the *Burrows-Wheeler Aligner* software (BWA, version 0.6.1) (Li & Durbin, 2009). We marked duplicate reads using *Picard* tools (v. 1.82, http://broadinstitute.github.io/picard) and sorted and indexed reads using *samtools* (v. 0.1.18) (Li *et al*, 2009). Using the *Genome Analysis Toolkit* (GATK, v. 2.3.4) (Depristo *et al*, 2011), we passed the data through various processing steps to realign and recalibrate the reads and to detect SNVs and short InDels; we followed the relevant GATK Best Practice Variant Detection guide.

On average per sample, 5,848,610,129 bases aligned to the reference genome, and 88.24% of bases fell within exome-capture target regions, resulting in a mean target coverage of 58.10. This level of coverage was considerably higher, for example, than the 20× coverage required for quality control in the TCGA breast cancer study (Koboldt *et al*, 2012).

Across all samples, we observed DNA variants at 941,507 unique loci (830,317 single nucleotide variants [SNVs] and 111,190 short insertion/deletion variants [InDels]). We used multiple criteria to filter the initial variants (Fig EV3). We excluded any variant for which a minor allele frequency greater than one percent (Cirulli & Goldstein, 2010) had been reported in any ethnic population in either the *1000 Genomes* (phase 1, release 3) (Abecasis *et al*, 2012) or *Exome Sequencing Project 6500* data (http://evs.gs.washington. edu/EVS).

Because variant calls are often discordant across sequencing technologies and analytical pipelines (O'Rawe *et al*, 2013), we used our variant-calling pipeline to process 611 germline DNA samples from The Cancer Genome Atlas (TCGA) that had been profiled via exome sequencing, and we excluded samples that had a minor allele frequency greater than 3%—a higher threshold was used for TCGA because this population may be enriched for susceptibility variants. Additionally, we excluded variants that occurred in more than 15% of the Utah samples but had not been excluded in prior steps. The preceding two steps reduced the number of SNVs and InDels by 18.4% and 90.8%, respectively.

Next, we excluded variants that fell outside exons (plus/minus two bases to allow for splice site mutations) used in each gene's primary transcript; gene/transcript definitions were extracted from *Entrez Gene*. The remaining variants were annotated for protein-coding effect using *snpEff* (Cingolani *et al*, 2012). We excluded variants that were assigned a severity level of "MODIFIER" or "LOW." We retained any variant that was assigned a "HIGH" severity; these consisted primarily of truncating, frameshift, and splice site variants. We also retained "MODERATE" InDels. We examined nonsynonymous coding SNVs for evolutionary conservation and functional effect using the SIFT (Kumar *et al*, 2009), Polyphen-2 (Adzhubei *et al*, 2013), and MutationAssessor (Reva *et al*, 2011) algorithms (obtained via dbNSFP (Liu *et al*, 2011) and aggregated them using Condel (Gonzalez-Perez *et al*, 2012)). We excluded any missense SNV called as "neutral." The remaining variants constituted our final set of "potentially pathogenic" variants. For simplicity, heterozygous variants and

homozygous rare variants were considered to have an equivalent effect; 99.5% of these variants were heterozygous.

After filtering, we observed 6,908 variants (average of 182.1 SNVs and 15.3 InDels per sample) at 5,551 loci. Most variants were non-synonymous substitutions (Appendix Fig S2). The most frequent substitutions were G-to-A and C-to-T transitions (Appendix Fig S3). The average transition/transversion ratio was 2.20. Most InDels resulted in a net gain/loss of three nucleotides or fewer; however, some were larger.

To assess the validity of our variant calls, we compared against the PCR-based, commercial genetic test results for *BRCA1* and *BRCA2*. We observed only one false-negative variant (rs80358061), which resides in a *BRCA1* intronic region outside the splice site junction points. Our pipeline identified this variant, but it was filtered out due to its intronic location. Five false-positive variants occurred, but in all cases except one, the false-positive variant coincided with another *BRCA1/2* variant in the same patient. Thus, for 94.2% of the samples, *BRCA1/2* mutation status was identified correctly via exome sequencing.

Although processing the TCGA data required substantial computational resources, we emphasize its importance. By processing these samples using the same pipeline that we used to process our own samples, we avoided systematic biases that can arise due to differences in variant-calling pipelines. Consequently, we identified variants/genes that were mutated frequently but that had not been identified in other databases we queried.

## Pathway-based analytic approaches

Using the PBMC gene expression data, we identified biological pathways that showed the greatest differences in expression between women who developed FBC and women who had a family history of breast cancer but who did not develop a tumor (Fig 1A). We obtained gene lists for 932 biological pathways from: (i) *KEGG* (Kanehisa *et al*, 2006) (accessed on June 16, 2011), (ii) the Molecular Signatures Database (v3.0) (Subramanian *et al*, 2005), and (iii) two research articles (Taube *et al*, 2010; Byers *et al*, 2013). For a given pathway, we made predictions in two successive steps: (i) we identified the most discriminatory genes from that pathway using the Support Vector Machines-Recursive Feature Elimination (SVM-RFE) algorithm (Guyon *et al*, 2002) and then (ii) used the SVM classification algorithm (Vapnik, 1998) to derive a probability that each patient had developed FBC. For the Utah individuals with a family history of breast cancer, we derived a probability for each sample in a ten-fold cross-validated design. We repeated this process for each pathway and ranked the pathways according to how accurately the SVM algorithm could distinguish women who developed FBC from women who had a family history of breast cancer but did not develop FBC ($n = 83$, see Table 1). We considered the pathways for which we attained the highest classification accuracy to be most likely to play a role in FBC development (Pang *et al*, 2006; MacNeil *et al*, 2015).

For the remaining samples, we estimated FBC status in a training/testing design. We trained an SVM model solely on the original 83 samples and predicted FBC status for the remaining samples, which included 60 Ontario women who had a family history of breast cancer, 28 of whom had developed breast cancer (Table 2). This set also included data for 54 women from Utah or Ontario who did not have a family history of breast cancer, 30 of whom had developed sporadic breast cancer. After ranking the genes via SVM-RFE (using only the training data), we derived SVM classification models for the top 25, 50, 75, 100, 125…300 ranked genes. The number of genes that performed best within the Utah data was then used for the training/testing analysis. The derived probabilities were then compared against FBC status, and an AUC value was calculated using the ROCR package (Sing *et al*, 2009). In this context, the AUC quantifies the model's ability to discriminate the groups at various probability thresholds; it can be interpreted as the frequency that the model would assign two randomly selected patients to the correct group. Finally, we derived a *P*-value for each pathway by comparing the AUC observed for that pathway against AUCs observed after randomly shuffling the class labels (1,000 permutations); these empirical *P*-values represent the fraction of permuted AUCs higher than the non-permuted AUCs.

We filtered these results further by excluding pathways for which the SVM predictions differed significantly (two sample *t*-test, *P*-value < 0.05) between (i) individuals who had a family history of breast cancer but did not develop a tumor and (ii) individuals who did not have a family history of breast cancer (irrespective of whether they developed a tumor). This filtering step helped to ensure that the pathway-level differences we observed were specific to FBC development and were not merely a result of treatment effects.

Subsequently, we applied the above approach to the Lim *et al* and Bellacosa *et al* data except that we used leave-one-out cross-validation within each data set. In addition, because we did not know which patients would eventually develop breast cancer, we compared patients with a family history of breast cancer and/or a BRCA1/2 variant against those who did not have these characteristics. Although it is not certain that individuals with a family history of breast cancer and/or a highly penetrant variant will eventually develop cancer, these individuals have a much higher risk than the remaining population (Stratton & Rahman, 2008).

For the DNA variant data, we aggregated variants at the gene and pathway levels. If a given sample carried any potentially pathogenic variant in a given gene or pathway, we considered that gene or pathway to be "mutated"; samples that contained a variant in a given gene carried an average of 1.02 variants in that gene. Under the assumption that frequently mutated genes are unlikely to drive susceptibility due to selective pressure and thus constitute noise at the pathway level, we excluded genes that are mutated relatively frequently. For this step, we used the germline samples from TCGA. Based on gene mutation frequencies in the TCGA data, we excluded genes from our data set that were mutated in more than 1.8% of TCGA germline samples (aside from *BRCA1/2*). We selected this threshold based on the maximal difference in the number of excluded genes for candidate thresholds that fell between 0.2% and 10% (Fig EV4). The number of "mutated" pathways was an average of 31 per patient (Fig EV3). We used a one-way Barnard's exact test (Barnard, 1945) to compare pathway-level mutation rates between individuals who developed FBC and those who did not. In cases where we sought to identify pathways whose mutation rates differed between BRCA1/2 and BRCAX individuals, we applied this test once in each direction.

When combining evidence across these various data sets, we used the *rankPvalue* function in the *WGCNA* package (Langfelder &

Horvath, 2008) to determine which pathways had a consistently low ranking, using *P*-value order within each data set for the rankings. Accordingly, although the *P*-value distributions differed across the analyses, the relative performance of each pathway was taken into account. We tested three alternative methods for combining *P*-values as implemented in the *metap* package (version 0.6; http://CRAN.R-project.org/package=metap). These included Fisher's combined probability test, Wilkinson's method, and the "sum p" method (Fisher, 1932; Wilkinson, 1951; Edgington, 1972). For each of these methods, we corrected for multiple tests using Storey's q-value method (version 2.0.0) (Storey, 2003).

### Correlation between sequencing variants and gene expression

For 34 samples that were profiled using both gene expression microarrays and DNA sequencing, we estimated the relationship between gene mutation status and expression of the same gene using Spearman's rank correlation coefficient. Using a local false discovery rate approach (Strimmer, 2008), we determined a threshold at which to reject the null hypothesis that gene mutation status was not correlated with expression levels of that gene. Fig EV5 illustrates the relationship between germline variant status and gene expression levels for 373 genes that showed the strongest association between these data types.

### Patient samples for *in vitro* assays

We collected additional primary mammary epithelial cells for 51 total patients from Utah. This cohort included reduction mammoplasty cells from women who lacked a family history of breast cancer and normal cells obtained via prophylactic surgery for women who had a family history of breast cancer. Family history status was determined via examination of medical health records. Different patients were used for each assay described below, due to limited amounts of tissue available after surgeries. Informed consent was obtained for all participants.

### Fluorescence microscopy

We examined cell morphology using fluorescence microscopy in a blinded manner. We seeded the first set of cells into multi-well slide chambers (Lab-Tek II CC2 Slide, 8 chambers) and grew them in MEBM basal media supplemented with a MEGM BulletKit (Lonza) for 3–5 days. We grew the second set of cells in STEMGENT WIT-P media for 1–2 days. Cells were fixed (3.7% formaldehyde, 15 min), permeabilized (0.5% Triton X-100, 5 min), and stained for F-actin (Alexa Fluor 568–phalloidin 1:150 [Molecular Probes]), focal adhesions (vinculin mouse antibody V-9131 1:1,000 (Sigma) with secondary antibodies Alexa Fluor 488–anti-mouse 1:200), and nuclei (Dapi 0.3 μM). Coverslips were mounted in Mowiol (Sigma). Cell images were captured with a Zeiss AxioCamMRm camera on a Zeiss Axioskop2 mot plus microscope (40× dry objective, 0.75 NA) using Zeiss AxioVision 4.8.1 software. Normal primary cells from each of 24 patient cell cultures were captured and analyzed in a blinded manner. Microscope images were processed using Adobe Photoshop v.8 and Illustrator CS v.11.

We used the *Fiji* (based on ImageJ from NIH) software (Schindelin *et al*, 2012) to quantify features within the fluorescence images.

First, we estimated cell size by calculating the percent of the field area covered by cells and dividing by the estimate number of nuclei per field. To identify the field area covered by cells, we converted the layered images to an 8-bit format, enhanced the color contrast, binarized, and then closed and filled features (see Computer Code EV1 for macro commands). We estimated the number of nuclei in Dapi fields using a similar approach but also applied the "Find Edges", "Watershed", and "Find Particles" commands. As a measure of cellular density, we calculated the Euclidean distance between the centers of each nucleus pair and identified the three shortest inter-nucleus distances in each field. Finally, we estimated the proportion of cell area covered by phalloidin or vinculin by subtracting background noise, enhancing, and binarizing the images that reflect these stains; we then divided these values by the overall estimated cell area per field. For each of the metrics, we evaluated differences between all observations across the groups using two-sided *t*-tests.

### Cell adhesion assays

We assessed the cells' adhesiveness to the extracellular membrane *in vitro*. We pre-coated 96-well plates overnight with 1 μg/ml human laminin (Millipore), which were washed in PBS prior to cell plating. We plated 3,000 cells from single-cell suspensions per well, and the average number of viable cells adhering to the plate after three hours across the replicates was compared against total cell counts of viable cells after 14 h, using the CellTiter-Glo Luminescent Cell Viability Assay (Promega). We calculated all *P*-values using a two-sided *t*-test.

### Drug-response assays

The FAK inhibitor PF573228 was purchased from Selleckchem and dissolved in 100% DMSO to generate 10 mM stock solutions of each. Stock solutions were stored at −80°C in aliquots. Human primary mammary epithelial cells were grown in MEBM basal media supplemented with MEGM Bullet (Lonza, NJ, USA). We seeded 4,000 cells per well for each patient sample in 100 μl of media and in 96-well plates. Cells for a total of 31 patients (15 from prophylactic surgeries and 16 from breast reduction patients) were seeded. After 24 h, the cells were treated with PF573228 (0.16–20 μM/ml), and viability was quantified after 96 h using MTT (Sigma-Aldrich, MO, USA). We calculated $EC_{50}$ values for each patient sample using a nonlinear regression line fitted to drug response in GraphPad. Negative values were converted to zero prior to EC50 calculation.

Gefitinib and afatinib, both inhibitors of growth factor receptors, were used in additional cell-based assays to test for differential response between cells from women who had a family history of breast cancer and those who did not. Specifically, we plated 1500 cells/well for each patient sample in 40 μl of MEBM basal media supplemented with a MEGM Bulletkit (Lonza), in half-well 96-well plates (Nunc), in triplicate for each drug dose. After 24 h, we added afatinib and gefitinib. We chose the indicated doses within the average linear range of drug effect: afatinib (10 nM–1 mM) and gefitinib (0.001–0.5 mM). We used a BIOMEK 3000 (Beckman Coulter, Brea, CA, USA) robot to seed the cells and dispense the drugs. After 96 h, cell viability was quantified using

the CellTiter-Glo Luminescent Cell Viability Assay (Promega). We calculated the proportion of viable cells for each dosage by comparing against cell counts for non-treated cells. We then calculated a summary value for each cell line as the slope of a linear regression line fitted to the cell count proportions.

## Western blot assays

Protein was extracted from either breast tissue chunks or pleural effusions. DEB lysis buffer (1% Triton X-100, 150 mM NaCl, 50 mM Tris pH 8.0, 5 mM EDTA, 0.1%SDS) with protease (Roche Complete Mini) and phosphatase inhibitor (PhosSTOP, Roche) was used for cell/tissue lysis. A Bradford assay was performed to quantify protein content in each sample. Thirty micrograms of protein was loaded onto 4–12% gradient gels (Bio-Rad) for separation by SDS–PAGE. As there was a large number of samples, three gels were run containing both FBC and control patient samples. For an internal control, two control patient samples were repeated in each gel. Protein was transferred to PVDF membranes, blocked with Superblock (Thermo scientific), and incubated with the following primary antibodies: β-actin (Cell Signaling #3700, 1:2,000), F-actin (Abcam #ab205, 1:500), vitronectin (Abcam #ab45139, 1:500), Integrin $\alpha$4 (Cell Signaling #8440, 1:500), Integrin $\alpha$5 (Cell Signaling #4705, 1:500), Integrin $\alpha$6 (Cell Signaling #3750, 1:500), ICAM2 (Cell Signaling #13355, 1:500), p53 (Cell Signaling #2527, 1:500), PTEN (Cell Signaling #9552, 1:500), and FAK (Cell Signaling #13009, 1:1,000). Horseradish peroxidase that linked whole antibody from donkey (GE Healthcare NA934V) and sheep (GE Healthcare NXA931) was used as the secondary antibody. Prior to film exposure, blots were incubated with Supersignal West Dura Extended Duration Substrate (Thermo Scientific) for 5 min. Blots were stripped using Restore PLUS Western Blot Stripping Buffer (Thermo Scientific) for re-probing. Band quantification was done using the *ImageJ* software. The expression of proteins was compared with the expression of loading control (β-actin).

## Software

We used the *Weka* software package (*SVMAttributeEval* module) (Hall *et al*, 2009) to execute SVM-RFE. We configured it to remove 10% of genes in each iteration and to remove a single gene per iteration when < 1% of genes remained. Otherwise, we used default configuration settings. We used the *e1071* R package (http://cran.r-project.org/package = e1071) and *LIBSVM* library (Chang & Lin, 2011) for SVM classification. In deriving the models, we selected the radial basis function kernel and used nested cross-validation to tune the C parameter. We used ML-Flex (Piccolo & Frey, 2012) to execute these steps in parallel. We used the *pROC* package to calculate 95% confidence intervals for AUC values (Robin *et al*, 2011).

We used the Python programming language (http://www.python.org) to parse and summarize data files, and a pre-release version of the SCAN software for microarray normalization.

Software scripts and code that were used to preprocess data and to filter and rank pathways can be accessed at https://github.com/srp33/BCRiskPathways. Computer Code EV2 contains a Jupyter notebook with code and data that were used to identify pathways of interest and to create manuscript figures.

## Data availability

Gene expression data are posted in *Gene Expression Omnibus* under accession number GSE47862. DNA variants are posted in the Database of Genotypes and Phenotypes under accession number phs001044.v1.p1.

**Expanded View** for this article is available online.

## Acknowledgements

We express sincere gratitude to the study participants who made this project possible. We acknowledge Patricia Bild, Mary Johnson, and Margarith Moos for inspiring this research. We thank Drs. Robert B. Weiss and Kenneth Boucher for recommendations on genetic and statistical analyses and Dr. Laurie K. Jackson for experimental assistance. We recognize allocations of computer time from the Fulton Supercomputing Lab at Brigham Young University and University of Utah's Center for High Performance Computing. Patient samples for the cell-based assays were contributed by the Breast Disease Oriented Team at the University of Utah. We thank Dr. Holly Dressman for advice and processing of microarray samples. Funding: AHB acknowledges R01GM085601, institutional funds, and a private donor for support of this project. SRP was supported by NIH training grant 5T32CA093247. WEJ and SRP acknowledge support from 1R01HG005692. JRM acknowledges support from the National Institutes of Health, UO1 CA084955. The Utah Breast Cancer Family Registry was supported through cooperative agreement from the National Institutes of Health U01CA69446, the National Center for Research Resources, and the National Center for Advancing Translational Sciences, National Institutes of Health grant UL1RR025764, and by award number P30CA042014 from the National Cancer Institute. This work was supported by grant UM1 CA164920 from the National Cancer Institute. The content of this manuscript does not necessarily reflect the views or policies of the National Cancer Institute or any of the collaborating centers in the Breast Cancer Family Registry (BCFR), nor does mention of trade names, commercial products, or organizations imply endorsement by the US Government or the BCFR.

## Author contributions

AHB, WEJ, and SRP were responsible for project planning and data analysis. AHB and SSB conceived the study. SRP executed computational analyses. AHB, LMH, and GS performed experimental work and analysis. PJM, ALC, MCB, and WEJ provided intellectual input on study design. AHB, ILA, SSB, ALC, TC, AES, LAN, JRM, and CAA were responsible for acquiring clinical and genomic data used in the study. AHB and SRP wrote the manuscript. LMH, ALC, LAN, AES, PJM, and SSB provided intellectual input on the manuscript.

## Conflict of interest

The authors declare that they have no conflict of interest.

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
