## [Review Process File · Molecular Systems Biology]

Integrative analyses reveal signaling pathways underlying familial breast cancer susceptibility

Stephen R. Piccolo, Laura M. Hoffman, Thomas Conner, Gajendra Shrestha, Adam L. Cohen, Jeffrey R. Marks, Leigh A. Neumayer, Cori A. Agarwal, Mary C. Beckerle, Irene L. Andrulis, Avrum E. Spira, Philip J. Moos, Sandra S. Buys, W. Evan Johnson and Andrea H. Bild

Corresponding author: Andrea Bild, University of Utah

Review timeline:

Submission date:	26 September 2014
Editorial Decision:	13 November 2014
Resubmission:	13 August 2015
Editorial Decision:	01 October 2015
Revision received:	30 December 2015
Editorial Decision:	08 February 2016
Revision received:	09 February 2016
Accepted:	11 February 2016

Editor: Maria Polychronidou

Transaction Report:

1st Editorial Decision

13 November 2014

Thank you again for submitting your work to Molecular Systems Biology. We have now heard back from the three referees whom we asked to evaluate your manuscript. As you will see from the reports below, the referees raise substantial concerns on your work, which, I am afraid to say, preclude its publication in Molecular Systems Biology.

While the reviewers appreciate that the presented integrative approach seems potentially interesting, they raise significant concerns regarding the conclusiveness of the study and they are not convinced that as it stands the suggested role of cell-adhesion pathway deregulation in breast cancer development is well supported. The reviewers rated the validity of the main conclusions as "low" and indicated that they would not support publication of this work in Molecular Systems Biology. Considering the substantial concerns and the overall low level of support provided by the referees, we see no other choice but to return this manuscript with the message that we cannot offer to publish it.

Nevertheless, as the reviewers did have positive words for the goals and the potential relevance of the study, we would not be opposed to considering a new and extended manuscript based on this work, provided that the most substantial points raised by the reviewers can be addressed convincingly. Specifically, a new submission should address the following points with the appropriate experimentation and analyses:

- The results need to be validated using a separate patient dataset.
- Additional analyses are required in order to better support the involvement of perturbed cell adhesion activity in the increased risk of breast tumors in FBC women. Reviewer #2 provides

constructive suggestions regarding this point. Since, as mentioned by reviewer #1 a link between genetic variants in cell adhesion pathway genes and breast cancer risk has been previously reported, providing further functional and/or mechanistic insights is going to significantly enhance the impact of the study.

- Samples positive or negative for BRCA1/2 mutations need to be analyzed separately.
All other issues raised by the referees would also need to be rigorously addressed.

A resubmitted work would have a new number and receipt date. We recognize that this would involve substantial additional experimentation and analyses with unclear outcome and, as you probably understand, we can give no guarantee about its eventual acceptability. Therefore we would understand if you would decide to submit the manuscript elsewhere.

If you do decide to follow this course then it would be helpful to enclose with your re-submission an account of how the work has been altered in response to the points raised in the present review.

REFEREE COMMENTS

Reviewer #1:

Piccolo and colleagues used an integrative genomics approach to identify novel mechanisms involved in familial breast cancer development. Specifically, the authors' primary approach was to derive perturbed signaling pathways in a cohort of women that developed familial breast cancer based on blood-derived gene expression measurements as well as germline exome sequencing information. Overrepresented signaling pathways were validated using external breast tissue gene expression measurements. Functional experiments were performed to further characterize deregulation of cell adhesion pathways in breast tissue cultures derived from high-risk women. The authors' computational and experimental results suggest that women with familial history of breast cancer exhibit decreased cell adhesion properties in normal breast tissue and that these perturbed molecular processes are in part driven by germline genetic variation in protein-coding genes.

Overall, this study provides further evidence to previous observations that (i) blood-derived molecular signatures harbor specific information about deregulated molecular processes in breast tissue (Sharma et al. *Breast Cancer Res*, 2005; Aaroe et al. *Breast Cancer Res*, 2010; Tudoran et al. *PLOS ONE* 2014), (ii) that deregulated cell adhesion pathway genes can be detected in blood cells of breast cancer patients (Tudoran et al. *PLOS ONE* 2014), and that (iii) germline variants in cell adhesion pathway genes contribute to breast cancer risk (e.g. Ayala et al. *Breast Cancer Res Treat* 2003; Langsenlehner et al. *Breast Cancer Res Treat* 2006; Liu et al, *Mol Biol Rep*, 2013). The analysis and experimental follow-ups are well done - a central question though is whether the advances described in this study, which come from the integrated approach used, are deemed of sufficient novelty for *Molecular Systems Biology* as they are indeed to an extent incremental (see comments further below) - I am not fully convinced currently that the extent of novelty is sufficient. Further comments are below:

Comments:

1) The Methods section states that among all participants "[...] 32.0% carried a known pathogenic mutation in BRCA1 or BRCA2 [...]" and that genetic variants in BRCA1/2 have been compared against a "PCR-based commercial genetic test". The authors should provide more information about how they have evaluated germline variants in BRCA1/2, i.e. which pathogenic germline variants in BRCA1/2 are known and which commercial genetic tests were used to evaluate BRCA1/2 mutation status.

2) Are germline variants in cell adhesion pathways genes specifically enriched in women that are negative for pathogenic BRCA1/2 mutations? This would provide further evidence that BRCA1/2-independent mechanisms might contribute to breast cancer risk.

- 3) It is not evident how ref. 20 supports the following statement in the introduction: "expression patterns in lymphoblastoid cells are distinct from those in individuals who do not carry these [BRCA1/2] mutations". Please clarify.
- 4) The authors removed genes that correlated with lymphocyte markers, epidemiological, demographic, and health factors, yet no details about this analysis were provided in the (Supplementary) Methods section. Please specify the exact regression factors (e.g. which lymphocyte markers), the statistical models, and the total number of excluded genes.
- 5) Figures 2B and 2C demonstrate that only 3 out of 49 (~6%) genes are shared between the gene expression- and germline variation-based pathway enrichment analysis. The authors should comment on this observation and discuss alternative mechanisms that might explain this discrepancy (e.g. pathogenic regulatory germline variation).
- 6) The significance of the results in Figure S3 is not clear. Please clarify in more detail the results of this analysis.
- 7) The authors should describe their findings in light of previous reports decreasing the novelty of this present study, in particular on
- a. Blood-derived molecular signatures and breast cancer (Sharma et al. Breast Cancer Res, 2005; Aaroe et al. Breast Cancer Res, 2010; Tudoran et al. PLOS ONE 2014)
 - b. Germline variation in cell adhesion pathway genes and breast cancer (e.g. Ayala et al. Breast Cancer Res Treat 2003; Langsenlehner et al. Breast Cancer Res Treat 2006; Liu et al, Mol Biol Rep, 2013)

Reviewer #2:

Bild and colleagues attempt to determine if there are systematic differences between transcriptional profiles AND SNVS from women at high risk for familial breast cancer. This could be a very important study. However, the data as presented is not convincing and further the lack of an independent test and validation set approach without concerns multiple testing is a major limitation to this paper.

1. Figure 2 B. The Gene expression analysis seems to have been done on all FBC patients without separating BRCA1/2 mutant and BRCAX cases. In Figure 2C, the SNV analysis is appropriately separated into BRCA1/2 and BRCAX patients. Importantly, the frequency of SNVs appears to be markedly different in these cases. This argues that the expression pattern markers should be different between BRCA1/2 and BRCAX cases. Indeed, given the effects of BRCA1/2 and also that these are tumor suppressors where the consequences of haploinsufficiency and reduction to homozygosity are likely to be very different, conflating BRCA1/2 and BRCAX cases does not seem appropriate. What are the gene expression predictors when the two groups are analyzed separately.
2. From figure 2, it is clear that while the Ontario and Utah sets were not mixed for discovery, they were not used in a training and test set approach. While the use of pathways does to a degree decrease the problems with the multiple comparisons problem, they are still many pathways tested against a limited number of patients. The authors need to test the results in a separate set of patients without a multiple comparisons approach.
3. In figure 2, the authors assess lymphocytes. The expression of multiple molecules that are involved in adhesion in epithelial cells are predicted to associate with development of breast cancer in the analysis of the lymphocytes. What were the relative levels of expression of these genes (preferably present this in comparison to those in the epithelial breast cells assessed later in the manuscript). Where they sufficient to allow adequate discrimination of pathway activation.
4. In figure 2C, what is the X scale. Is this number of cases of a SNV in the gene. As a specific point in the description of figure 2, the authors claim: "Together, these findings suggest that genetic variation and aberrant transcriptional activity (in part driven by genetic variation, see Figure S3) perturb normal cell adhesion activity, leading to an increased risk of breast tumors in FBC women." This statement claims a causal role for what at this point in the paper is at best an association that requires further validation. Please correct these and similar claims for causality from association studies.

5. In figure 2C what is the statistical approach used to determine whether the association with SNVs is significant. What genes were tested to find the set in figure 2C. Was this genome wide, all genes in the integrin and small cell lung cancer pathways? What was the correction used for the multiple comparisons problem.
6. The Lim and Bellacosa et al sets provide test sets. However, it is not clear how many different predictors were tested on these sets and importantly whether a Bonferonni correction was done for multiple testing. Indeed in the supplemental data file 3, it appears that a large number of predictors were assessed on these sets. In the text it states "When these data sets were included in our analysis, twelve pathways remained significant (p value < 0.05)." p is <0.05 is not appropriate where there is multiple testing. What correction approach was used?
7. A key set of experiments that could provide validation relates to the assessment of patients undergoing breast reduction or prophylactic surgery. It is important to emphasize that the surgical procedures used in these two operations are remarkably different in degree and also in terms of length of surgery. Further, there may be markedly different cold ischemic times while pathology evaluates the specimen. What was done to prevent potential biases. At what stage of the analysis was the person performing the analysis blinded. This is critical from the pathologist involved through to the person assessing the slides.
8. The authors should assess the same adhesion markers that were predictors in the transcriptional analysis in the cultured cells. While cell area, phalloidin and vinculin are interesting markers, they are not the markers predicted by the analysis in figure 2. The authors need to explain what E Cadherin and Vimentin are not altered as they would be expected to be altered in cells with marked alterations in cell cell adhesions.
9. The potential markers were assessed in cells in culture for a significant period of time. This is a marked perturbation. The authors need to assess the adhesion markers by IHC in freshly collected sample that had not been cultured to demonstrate that the differences actually occur in vivo.
10. The adhesion studies are very interesting and supportive. There is a cryptic comment about time dependent changes in the text. These need to extended, presented and then compared to the studies in figure 4 and other fluorescent assays.
11. The studies with TRAIL, and other inhibitors is intriguing. However, linking trail sensitivity to decreased adhesion is a major leap. What is the data that the increased TRAIL sensitivity is indeed due to differential adhesion characteristics.

Reviewer #3:

"Novel signaling pathways underlying familial breast cancer susceptibility"

General comment:

The paper presents an interesting framework to summarise genomic data at the pathway level. Using gene expression data from two cohorts the authors identify multiple cell-adhesion pathways de-regulated in high risk patients. The findings were validated experimentally by comparing cell cultures from patients with and without family history of breast cancer. The authors suggest that pathway disruption obtained from profiling peripheral blood cell may play a role in tumorigenesis.

Overall, the computational methodology applied here is interesting and can be applied to other cancers with the same type of available data. However there isn't much detail on the biological implications of the findings. There is no mechanistic insight to how the disruption of cell-adhesion pathways may influence breast cancer development. There is also no detail about what different subtypes of familial breast cancer were considered in this study.

Although the authors present a innovative and clever way of integrating multi-omics datasets there is no sufficient data to support disease-related mechanisms. In addition, some statements regarding methods and conclusions were too vague and need to be addressed:

The authors should address the following issues:

1 - Abstract

"Both genomic and functional experiments support the concept that cell-cell and cell-extracellular matrix adhesion processes are disrupted in non-malignant cells of many women who have a family history of breast cancer and that these processes play a role in FBC development."

The authors report an association between high-risk and pathway disruption. But this association does not support the concept that pathway disruption plays a role in breast cancer development. This sentence should be rephrased (for example: "these processes may play a role in FBC development).

2 - Need of a table summarising the information for each cohort (Table S1 is too simplified).

The authors analysed 2 cohorts: Utah (124 women) and Ontario (73 women).

In addition to table S1, there should be a table with better description of all data in each cohort.

The contain the following information:

- How many individuals were grouped into the 6 groups (No FH with and without cancer, BRCA1/2 with and without cancer, BRCA1/2 with and without cancer).

- For those groups "with cancer", what was the subtype of familial cancer (ER-, HER2-, etc) and what grade?

- For those groups "with family history" - how many first-degree relatives have been diagnosed with breast cancer? what subtype of breast cancer (ER-, HER2-, etc)?

- What is the average age per group (ok in figure S1 but should add this info to the table and for both cohorts)

- Which cases were used as controls? Authors say there were 41 from Utah and 13 from Ontario, with no family history of breast cancer. But it is not very clear which group they refer too.

3 - More concise explanation of the cohorts used in this study.

The authors say " Where possible, patients were matched according to these criteria and by age at which blood was drawn. (...) cohorts included women who either 1) had a family history of breast cancer or did not, 2) carried a pathogenic germline variant in BRCA1 or BRCA2 or did not, and 3) had developed breast cancer or had not"

But no table is provided showing exactly all available information for all individuals considered in this study.

I find this section of the methods on the study design not very clear. A table summarising clearly what data was used and how the groups were built will help organising this information.

4 - too vague

"Individuals who had not developed breast cancer or who did not have a family history of breast cancer served as controls." (page 6)

How many individuals were controls? Too vague.

5 - not very explicit

"The samples showed no statistical difference among all groups in covariates, including age." (page 6)

This statement is too vague for a "methods" section. What covariates were tested and how? please explain

6 - more detail needed

"We excluded any gene whose expression correlated with lymphocyte markers or epidemiologic/demographic/health factors"

Not sure what this means and how this analysis was made. What were the epidemiologic/demographic/health factors considered and what do you mean with "correlated"?

7 - reference needed

"We then used the Support Vector Machines (SVM) algorithm to identify multigene patterns that significantly differed between the patient groups." (page 8)

Add citation (was it in R? what package?)

For all data analysis in R, please provide Rmarkdown or knitR files to reproduce your results.

8 - link leads to too many undocumented data files

"All software scripts and code that were used to execute these analyses can be accessed at <https://github.com/srp33/BCSP>" (page 9)

In this link there are 9 folders, one of them is called "scripts". The "scripts" folder contains 21 files of (I guess) scripts. The names of the files are "preparefastq", "prepend_string_to_file", "decompress", "delete_temp_files", etc. There is another subfolder in "scripts", with an extra 20 data files.

The idea of making the research reproducible by sharing the code is a great one. The authors could also include some documentation to explain the user what folder/file(s) they should run to reproduce a each section of the paper.

9 - Some figures need to explicitly say what was used as a "control":

- In figure 3 and 5 what are the "controls"?

(For example, for figure 3, are "controls" women who did not had a family history of breast cancer?)

Add this information to both Figure 3 and Figure 5 caption to be more explicit.

10 - Discussion

Instead of "Together our findings suggest that aberrant signaling within cancer pathways in peripheral blood may be a sign of eventual tumor development in general." (page 17)

A bit too vague. Instead of "a sign of eventual tumor development in general", the authors should say more clearly that disruption of cell adhesion pathways is associated to higher risk of developing familial breast cancer.

Resubmission

13 August 2015

Summary of Dr. Polychronidou's comments: *While the reviewers appreciate that the presented integrative approach seems potentially interesting, they raise significant concerns regarding the conclusiveness of the study and they are not convinced that as it stands the suggested role of cell-adhesion pathway deregulation in breast cancer development is well supported.... [n]evertheless, as the reviewers did have positive words for the goals and the potential relevance of the study, we would not be opposed to considering a new and extended manuscript based on this work, provided that the most substantial points raised by the reviewers can be addressed convincingly. Specifically, a new submission should address the following points with the appropriate experimentation and analyses:*

(1)- *The results need to be validated using a separate patient dataset.*

Response to (1): As described below in our response to Reviewer #2, we further detail the validation of our findings in multiple patient data sets. In particular, we emphasize that we performed our PBMC-based, pathway analysis using a training and independent validation testing design. Specifically, we trained the algorithm using data from our Utah cohort only and then applied them independently to the remaining samples. Therefore, with this approach all predictions for our test set were derived solely from Utah samples, thus illustrating that our findings are consistent across these independent populations. We have improved the wording in the manuscript to make this clear. In addition, we have added Tables S2-S3, which contain additional information about these cohorts. Our paper also describes follow-up validations on two additional and completely separate gene-expression data sets of normal breast tissue, providing another level of validation in addition to the Ontario cohorts. Further, in the lab, we have tested our findings on two additional cohorts of women from Utah who donated breast tissue for our cell-based analyses, as requested by the reviewers. Again, these cohorts are distinct from the original cohorts. In all, this study focuses on findings that were consistent across all of these patient sets, and both computational and biochemical studies.

(2)- *Additional analyses are required in order to better support the involvement of perturbed cell adhesion activity in the increased risk of breast tumors in FBC women. Reviewer #2 provides constructive suggestions regarding this point. Since, as mentioned by reviewer #1 a link between genetic variants in cell adhesion pathway genes and breast cancer risk has been previously*

reported, providing further functional and/or mechanistic insights is going to significantly enhance the impact of the study.

Response to (2): In response to this concern, we have performed two additional assays to complement our cell adhesion and microscopy studies. As detailed in our response to reviewer #2, we have collected additional series of breast tissue samples from high-risk women and controls. With this additional series, we have performed an experiment in which we used a Focal Adhesion Kinase (FAK) inhibitor (PF573228) to show that primary breast epithelial cells from women who have a family history are more responsive to this drug than women who do not have a family history of breast cancer, suggesting that FAK signaling and modulation of cell adhesion is a mechanism of breast cancer risk in high risk women. We also used Western blots to profile expression levels for nine proteins that play a role in focal adhesion. In response to Reviewer #1, we have also updated the manuscript to place it in context with prior work in this area and to emphasize the key contributions of this study that build on prior work. We believe these contributions are substantial.

(3)- *Samples positive or negative for BRCA1/2 mutations need to be analyzed separately.*

Response to (3): Detailed below in response to reviewers #2 and #3, we have updated Figure 2 and performed an additional analysis, treating BRCA1/2 individuals separately from BRCAX individuals. The results from these studies are consistent with our conclusions, and show that they are independent of BRCA1/2 status.

(4)- *All other issues raised by the referees would also need to be rigorously addressed.*

Response to (4): We have addressed all other issues raised by reviewers below in a point-by-point format.

Reviewer #1:

(Summary) *Reviewer #1 observes that our research “provides further evidence to previous observations” that blood-derived molecular signatures harbor specific information about deregulated cell adhesion molecular processes in breast tissue. We thank the reviewer for detailing that our “analysis and experimental follow-ups are well done”. However, the reviewer wanted more detail on how the “advances described in this study, which come from the integrated approach used, are deemed of sufficient novelty”.*

We thank the reviewer for taking time to review our work thoroughly and for providing such helpful comments. We address the reviewer’s summary of concerns here, along with a point-by-point response to the reviewers’ comments below. Our study is unique in that we developed a computational, pathway-based “pipeline” to identify signaling pathways that distinguish high-risk affected women from high-risk unaffected women. Therefore, our study is unique in the following two important ways:

1. We are studying pathways versus single genes, using multiple types of genomic data (gene expression and DNA mutations).
2. We are studying pathways that distinguish high-risk women who developed breast cancer from those who did not. This approach is in contrast to others that have investigated individual susceptibility genes.

As the reviewer detailed in her/his summary, previous efforts have studied individual genes that are important in breast cancer risk, independent of high-risk status. Our studies are, in our opinion, a unique approach as we are identifying pathways that are dysregulated in high-risk women who develop breast cancer versus high-risk women who do not develop breast cancer. To further detail these differences, we provide detailed clarifications below for the papers that the reviewer referenced. We now cite these papers and below emphasize various ways in which our paper extends beyond what was performed in those studies.

Specific references from reviewer:

- i) *Reviewer: Blood-derived molecular signatures harbor specific information about deregulated molecular processes in breast tissue (Sharma et al. Breast Cancer Res, 2005; Aaroe et al. Breast Cancer Res, 2010; Tudoran et al. PLOS ONE 2014)*

- a. Response: These manuscripts (Sharma et al. Breast Cancer Res, 2005; Aaroe et al. Breast Cancer Res, 2010) detail biomarkers of early detection of breast cancer, and are not specific for familial breast cancer. The Tudoran et al. PLOS ONE 2014 paper distinguishes HER2+ breast cancer using peripheral blood in sporadic cancer patients, and does not interrogate cell adhesion pathway-level data and is not applied in the context of familial breast cancer.
- ii) Reviewer: *That deregulated cell adhesion pathway genes can be detected in blood cells of breast cancer patients (Tudoran et al. PLOS ONE 2014).*
- a. Response: Again, this paper is applied in a different context (sporadic HER2+ breast cancer and not familial breast cancer) and is focused on various single genes and is not pathway based.
- iii) Reviewer: *Germline variants in cell adhesion pathway genes contribute to breast cancer risk (e.g. Ayala et al. Breast Cancer Res Treat 2003; Langsenlehner et al. Breast Cancer Res Treat 2006 (this paper details the COX2 gene, not adhesion pathways); Liu et al, Mol Biol Rep, 2013).*
- a. Response: These papers investigate mutations only (not expression) of breast cancer susceptibility genes that distinguish normal versus women affected with breast cancer in case and control cohorts, and not the pathways that distinguish within high-risk women who develops familial breast cancer versus those high-risk women that do not.

We hope that we have clarified the uniqueness of our approach and our results, and we apologize this was not clear before. We have made edits to the text to clarify these points.

In addition to the reviewer summary, below we respond to reviewer comments point-by-point.

Comments:

1) *The Methods section states that among all participants "[...] 32.0% carried a known pathogenic mutation in BRCA1 or BRCA2 [...]" and that genetic variants in BRCA1/2 have been compared against a "PCR-based commercial genetic test". The authors should provide more information about how they have evaluated germline variants in BRCA1/2, i.e. which pathogenic germline variants in BRCA1/2 are known and which commercial genetic tests were used to evaluate BRCA1/2 mutation status.*

We have added detail to the manuscript about how BRCA1/2 mutation status was identified (genetic testing at Myriad Genetics). This includes a link to a document (<https://www.myriad.com/lib/technical-specifications/BRACAnalysis-Technical-Specifications.pdf>) that describes these tests. For the subset of patients for whom we performed exome sequencing, we also indicate the exact mutations that we observed (Supplementary Data File 1) as well as the process we used to estimate pathogenicity. As we indicate in the manuscript, our exome-sequencing results correlate strongly with the results of this commercial test.

2) *Are germline variants in cell adhesion pathways genes specifically enriched in women that are negative for pathogenic BRCA1/2 mutations? This would provide further evidence that BRCA1/2-independent mechanisms might contribute to breast cancer risk.*

To evaluate whether our top pathways were enriched specifically in BRCAX women who developed breast cancer, we compared pathway-level mutation rates between 1) BRCAX individuals who developed breast cancer and 2) BRCA1/2 individuals who developed breast cancer. Pathway-level mutation rates differed significantly ($p < 0.05$) for only 2 of the 45 pathways we highlighted in the PBMC analysis, suggesting that the pathways we identified are broadly relevant to familial breast cancer. Future research with larger cohorts will help to better decipher the interplay among these mechanisms.

3) *It is not evident how ref. 20 supports the following statement in the introduction: "expression patterns in lymphoblastoid cells are distinct from those in individuals who do not carry these [BRCA1/2] mutations". Please clarify.*

The reviewer is correct. This reference was mixed up with another, thank you for this correction. We have fixed this error in the revised manuscript.

4) *The authors removed genes that correlated with lymphocyte markers, epidemiological, demographic, and health factors, yet no details about this analysis were provided in the (Supplementary) Methods section. Please specify the exact regression factors (e.g. which lymphocyte markers), the statistical models, and the total number of excluded genes.*

We have added much more detail to the manuscript about how the gene-expression data were normalized, summarized, and filtered. This includes a detailed statement about the regression factors, statistical approach, and number of genes that were excluded. Below is the statement specifically about the latter:

“We also sought to identify and filter out any genes whose expression might be confounded by immune activity or demographic/clinical variables associated with the patients. We applied a total lymphocyte enumeration test to whole blood cells for 22 samples from the Utah cohort to estimate the number of B-cell, CD3-positive T cells, CD4-positive T-cells, CD8-positive T-cells, and NK-cells. In addition, 63 patients from the Utah cohort responded to a health-assessment survey that collected the following variables for each participant: age at diagnosis, age of first menstrual cycle, time since last menstrual cycle, age when menstruation ceased, level of education, religious preference, overall health status, marital status, level of physical activity, use of contraceptives, total number of pregnancies, age at first live birth, age at last live birth, number of live births, breastfeeding status, use of chemopreventive/hypertension/anti-inflammatory drugs, tobacco use, alcohol use, occupational history, and history of immunological disorders. We used a multifactor ANCOVA test to identify genes ($n = 334$) for which gene-expression levels correlated strongly with any of these factors ($p < 0.01$).”

5) *Figures 2B and 2C demonstrate that only 3 out of 49 (~6%) genes are shared between the gene expression- and germline variation-based pathway enrichment analysis. The authors should comment on this observation and discuss alternative mechanisms that might explain this discrepancy (e.g. pathogenic regulatory germline variation).*

We thank the reviewer for this helpful comment. We have added the following statement to the Results section:

“As shown in Figures 2B-C, the individual genes within a given pathway that showed the strongest differences in expression levels among our patient groups often were not the same genes that showed significant differences in germline mutation frequency. Thus it is likely that mechanisms beyond rare germline variants influence peripheral blood expression levels for these pathways; such mechanisms might include common germline variants, epigenetic variation, and microRNA expression.”

6) *The significance of the results in Figure S3 is not clear. Please clarify in more detail the results of this analysis.*

We have added clarifying text both to the main manuscript (Results section) and to the caption for the figure (see below):

“We sought to identify genes whose expression levels correlated strongly with mutation status. This figure shows data for 373 genes that exhibited the strongest association between the presence of one or more potentially pathogenic variants and expression of the same gene. Red dots indicate samples that carried a mutation within a given gene. Gray dots indicate samples that did not carry a mutation within the same gene. This figure shows that in many cases, germline variants are correlated with a considerable increase or decrease in expression levels for the same gene. Expression values for each gene are standardized to a consistent scale for illustration purposes.”

7) *The authors should describe their findings in light of previous reports decreasing the novelty of this present study, in particular on*

a. Blood-derived molecular signatures and breast cancer (Sharma et al. Breast Cancer Res, 2005; Aaroe et al. Breast Cancer Res, 2010; Tudoran et al. PLOS ONE 2014)

b. Germline variation in cell adhesion pathway genes and breast cancer (e.g. Ayala et al. Breast Cancer Res Treat 2003; Langsenlehner et al. Breast Cancer Res Treat 2006; Liu et al, Mol Biol Rep, 2013)

We now refer to the studies that the reviewer mentioned, and we have clarified areas in which our manuscript makes novel contributions as detailed above. In addition, our analysis extends prior observations by aggregating evidence across genomic data from four independent patient populations (five data sets total)—including two that characterized expression within normal breast cells—and by assessing these observations in vitro using cell-based functional assays, drug-response assays, and fluorescence microscopy. Such integrative approaches are imperative for deciphering the mechanisms that influence disease risk.

Reviewer #2:

Bild and colleagues attempt to determine if there are systematic differences between transcriptional profiles AND SNVS from women at high risk for familial breast cancer. This could be a very important study. However, the data as presented is not convincing and further the lack of an independent test and validation set approach without concerns multiple testing is a major limitation to this paper.

We thank the reviewer for evaluating our work carefully and for the various comments and suggestions that have been provided. We address these comments and suggestions below. We further detail our multiple validation cohorts in points #2 and #6 below. Note, validation analyses were carried out both using genomic characterization of independent datasets as well as at the bench in functional/pharmacologic experiments.

1. Figure 2B. The Gene expression analysis seems to have been done on all FBC patients without separating BRCA1/2 mutant and BRCA1 cases. In Figure 2C, the SNV analysis is appropriately separated into BRCA1/2 and BRCA1 patients. Importantly, the frequency of SNVs appears to be markedly different in these cases. This argues that the expression pattern markers should be different between BRCA1/2 and BRCA1 cases. Indeed, given the effects of BRCA1/2 and also that these are tumor suppressors where the consequences of haploinsufficiency and reduction to homozygosity are likely to be very different, conflating BRCA1/2 and BRCA1 cases does not seem appropriate. What are the gene expression predictors when the two groups are analyzed separately.

We thank the reviewer for this comment. The main goal of our analysis was to identify pathways that differed consistently between individuals 1) who had a family history of breast cancer and developed a tumor and 2) those who had a family history of breast cancer but did not develop a tumor—irrespective of BRCA1/2 mutation status. As indicated by past work (Waddell N, 2008, et al.), we agree that BRCA1/2 mutations will likely affect PBMC expression levels in ways that are specific to individuals who carry those mutations. To address such effects, our control group included individuals who carried BRCA1/2 mutations but did not develop breast cancer. In response to the reviewer's comment, we have updated Figure 2B so that it separates expression levels according to BRCA1/2 mutation status as well as cancer-development status. As would be expected, in some cases, expression patterns for genes in these pathways differ between individuals who carry a BRCA1/2 mutation and those who do not. However, frequently we see similarities in expression among individuals who developed familial breast cancer—irrespective of BRCA1/2 status—and consistent differences between individuals who developed familial breast cancer and those who did not. We have updated these figures in the manuscript. Our objective was to identify these similarities for a more inclusive assessment of familial breast cancer risk, independent of mutation status in any one specific susceptibility gene.

2. From figure 2, it is clear that while the Ontario and Utah sets were not mixed for discovery, they were not used in a training and test set approach. While the use of pathways does to a degree decrease the problems with the multiple comparisons problem, they are still many pathways tested against a limited number of patients. The authors need to test the results in a separate set of patients without a multiple comparisons approach.

We did use a training and test set approach, and we have clarified this in the manuscript—our apologies if this approach was not clear previously. In the text, we state that, “First we tested this approach on the Utah cohort via 10-fold cross validation...For external validation, we used a training/testing design. We trained the SVM algorithm using data from the same Utah patients (n = 83) and derived predictions for the remaining “test” samples. The test set included data for 60 Ontario women who had a family history of breast cancer, 28 of whom had developed breast cancer (Table S3). The test set also included data for 54 women from Utah or Ontario who did not have a family history of breast cancer, 30 of whom had developed sporadic breast cancer (see Tables S2-S3).” We hope these edits clarify our approach with sufficient detail.

3. In figure 2, the authors assess lymphocytes. The expression of multiple molecules that are involved in adhesion in epithelial cells are predicted to associate with development of breast cancer in the analysis of the lymphocytes. What were the relative levels of expression of these genes (preferably present this in comparison to those in the epithelial breast cells assessed later in the manuscript). Were they sufficient to allow adequate discrimination of pathway activation.

We now provide Supplementary Data File 5, which indicates gene-expression values for the genes in the pathways that are represented in Figure 2. These values are provided for the Utah, Ontario, Lim, et al., and Bellacosa, et al. data sets. Thus our readers can easily evaluate the data values for each sample and make comparisons between what we observed for our PBMC samples and for the normal breast tissue samples (Lim, et al. and Bellacosa, et al.). In addition, we emphasize that these pathways performed consistently well in all of our pathway-level analyses (Figures 2-3 and Supplementary Data Files 3-4). These analyses took into account complex, multigene patterns and ensured that the patient groups could be discriminated effectively based on these patterns. Thus although relative expression levels may differ for individual genes across the data sets, our pathway-level findings suggest that these pathways are consistently dysregulated between the groups in both PBMC and breast tissue samples.

4. In figure 2C, what is the X scale. Is this number of cases of a SNV in the gene?

The black dots on Figure 2C represent individual patients samples that carried a likely pathogenic variant in one of the genes listed on the x axis. Each cell indicates whether a given patient carried one of these variants. We have clarified this in the figure caption.

As a specific point in the description of figure 2, the authors claim: "Together, these findings suggest that genetic variation and aberrant transcriptional activity (in part driven by genetic variation, see Figure S3) perturb normal cell adhesion activity, leading to an increased risk of breast tumors in FBC women." This statement claims a causal role for what at this point in the paper is at best an association that requires further validation. Please correct these and similar claims for causality from association studies.

We have altered this statement so that it claims a potential associative role rather than a causal role. We have also modified the wording in the Discussion section along these lines.

5. In figure 2C what is the statistical approach used to determine whether the association with SNVs is significant. What genes were tested to find the set in figure 2C. Was this genome wide, all genes in the integrin and small cell lung cancer pathways? What was the correction used for the multiple comparisons problem.

For a given pathway, we identified the number of samples that carried at least one likely pathogenic variant in any gene in the pathway. For samples that met these criteria, we considered the pathway to be “mutated” for those individuals. Then we used a one-way Barnard’s exact test to compare the number of samples for which the pathway was mutated between women who developed FBC and women who did not. Due to the relatively small sample number of samples we used for exome sequencing (n=35), we made no claims based on these observations alone. Rather, we focused on pathways that showed pathway-level differences not only for germline variants but also across the various gene-expression data sets we examined.

6. The Lim and Bellacosa et al sets provide test sets. However, it is not clear how many different predictors were tested on these sets and importantly whether a Bonferonni correction was done for

multiple testing. Indeed in the supplemental data file 3, it appears that a large number of predictors were assessed on these sets. In the text it states "When these data sets were included in our analysis, twelve pathways remained significant (p value < 0.05)." p is < 0.05 is not appropriate where there is multiple testing. What correction approach was used?

When we analyzed the Lim, et al. and Bellacosa, et al. data sets, we focused on the 45 pathways that fell below the 0.05 significance threshold in our earlier analysis, which was conducted on the Utah and Ontario data (see Supplementary Data File 3). For each of these pathways, we then evaluated pathway-level results for the two validation sets (Lim, et al. and Bellacosa, et al.). We aggregated all pathway results for each of these pathways and calculated a single p-value per pathway using a rank-aggregation approach (rankPvalue function within the WGCNA R package). Our approach focused on aggregating evidence across various data sets and conditions to identify the most robust and consistent signals in the data and then to validate those findings via experimental means. We focused on the 12 pathways that attained significance based on the following criteria: 1) rank-aggregation p-value < 0.05 , 2) Lim, et al. p-value < 0.05 , and 3) Bellacosa, et al. p-value < 0.05 . We used these stringent criteria to ensure that we focused only on pathways that performed well across all five datasets but also specifically on both validation datasets.

Requiring significance in multiple, independent validation studies (Lim and Bellacosa) is an effective way of conservatively controlling for multiple testing. For example, the probability that a null hypothesis will be falsely called significant by chance in two independent studies is $0.05 \times 0.05 = 0.0025$, or a 1-in-400 chance. A Bonferroni cutoff on 45 pathways/tests would require a p-value cutoff of $0.05/45 = 0.0011$ across both studies, or equivalently, would require a p-value threshold of less than 0.033 from both studies independently ($0.033^2 = 0.0011 = 0.05/45$). However, because these pathways are NOT independent (Bonferroni assumes independence), we are comfortable assuming that a Bonferroni cutoff is too conservative for this case. However, we still note that 8 of the 12 pathways would have passed this standard Bonferroni threshold, with the other 4 pathways falling just above the threshold (highest p-value across both studies 0.041). In addition, coupling this with our biological (lab-based) validation, we are confident that our conclusions are strongly justified as there is significant statistical and biological evidence (significance in 5 genomic datasets plus additional laboratory-based tests) that our observed phenomena are real effects.

7. A key set of experiments that could provide validation relates to the assessment of patients undergoing breast reduction or prophylactic surgery. It is important to emphasize that the surgical procedures used in these two operations are remarkably different in degree and also in terms of length of surgery. Further, there may be markedly different cold ischemic times while pathology evaluates the specimen. What was done to prevent potential biases. At what stage of the analysis was the person performing the analysis blinded. This is critical from the pathologist involved through to the person assessing the slides.

Dr. Leigh Neumayer, a surgeon and co-author on the paper, was consulted prior to study initiation. She put various controls in place to ensure that the time from surgery to sample collection was consistent, irrespective of surgery type. Upon evaluating this process, Dr. Neumayer verified that the cold ischemia time was less than 30 minutes. Also importantly, the experiments detailed in this paper used viable tissue collected from both breast reduction and prophylactic surgeries. These samples are processed as viable tissue, and grown in the same culture conditions for each experiment. Thus, the growth conditions for these samples were the same, and again minimize influence of collection effects. We have clarified these points in the manuscript.

8. The authors should assess the same adhesion markers that were predictors in the transcriptional analysis in the cultured cells. While cell area, phalloidin and vinculin are interesting markers, they are not the markers predicted by the analysis in figure 2. The authors need to explain what E Cadherin and Vimentin are not altered as they would be expected to be altered in cells with marked alterations in cell cell adhesions.

In response to the reviewers' recommendations, we have performed a Western Blot analysis to help validate the gene expression data that we show in Figure 2. The Western Blots were performed using an independently collected cohort of patient cells (please see Figure S7 and Table S3 in the manuscript). The reviewer is correct that some of the initial results presented in the manuscript reflected general cell adhesion properties and did not necessarily correspond to Figure 2. We have

clarified this with this additional analysis. Lastly, as the reviewer is correct that vimentin and E-cadherin are not markers listed in our genomic studies, and were instead intended to interrogate the EMT status of the patient cells, we have removed these specific experiments from the manuscript.

9. The potential markers were assessed in cells in culture for a significant period of time. This is a marked perturbation. The authors need to assess the adhesion markers by IHC in freshly collected sample that had not been cultured to demonstrate that the differences actually occur in vivo.

In order to address this comment, we have performed Western Blotting from snap frozen tissue as detailed in Figure S7 and Table S3.

10&11. The adhesion studies are very interesting and supportive. There is a cryptic comment about time dependent changes in the text. These need to be extended, presented and then compared to the studies in figure 4 and other fluorescent assays. The studies with TRAIL, and other inhibitors is intriguing. However, linking trail sensitivity to decreased adhesion is a major leap. What is the data that the increased TRAIL sensitivity is indeed due to differential adhesion characteristics.

We were able to collect additional independent patient samples for experiments that the reviewer requested. Specifically, we were able to request additional tissue collections from the collaborating surgeons and collected over 25 additional samples for evaluation. Given the challenging nature of collecting these samples and the limited tissue available for experiments, we focused on the Western Blotting analyses (Figure S7) and a dose-response assay. As the reviewer believes TRAIL sensitivity may not directly relate to adhesion properties, we used a cell-adhesion specific drug (Focal Adhesion Kinase inhibitor, PF573228) and excluded the TRAIL findings. We show that primary breast epithelial cells from women who have a family history are more responsive to the FAK inhibitor than women who do not have a family history of breast cancer (detailed in Figure 5). Together, these results strengthen our conclusion that proteins involved in cell adhesion show differential expression in high-risk women and that cell adhesion pathways are differentially activated between women who develop familial breast cancer and controls. We agree with the reviewer that additional studies of primary patient samples focusing on time dependent changes in cell adhesion/shape in culture would be informative; however, with limited tissue available, we cannot perform that experiment at this time. Therefore, we have removed the time-dependent statement from the text, and will pursue those experiments once we have again accrued sufficient patient samples.

Reviewer #3:

General comment:

The paper presents an interesting framework to summarise genomic data at the pathway level. Using gene expression data from two cohorts the authors identify multiple cell-adhesion pathways de-regulated in high risk patients. The findings were validated experimentally by comparing cell cultures from patients with and without family history of breast cancer. The authors suggest that pathway disruption obtained from profiling peripheral blood cell may play a role in tumorigenesis.

Overall, the computational methodology applied here is interesting and can be applied to other cancers with the same type of available data. However there isn't much detail on the biological implications of the findings. There is no mechanistic insight to how the disruption of cell-adhesion pathways may influence breast cancer development. There is also no detail about what different subtypes of familial breast cancer were considered in this study.

Although the authors present a innovative and clever way of integrating multi-omics datasets there is no sufficient data to support disease-related mechanisms. In addition, some statements regarding methods and conclusions were too vague and need to be addressed.

We thank the reviewer for a thorough review and for these positive comments. We have addressed the reviewer's concerns below, including additional studies (detailed above and in the revised manuscript) in pinpointing FAK as an important signaling pathway in FBC, and we believe these revisions strengthen our research.

The authors should address the following issues:

1 - Abstract

"Both genomic and functional experiments support the concept that cell-cell and cell-extracellular matrix adhesion processes are disrupted in non-malignant cells of many women who have a family history of breast cancer and that these processes play a role in FBC development."

The authors report an association between high-risk and pathway disruption. But this association does not support the concept that pathway disruption plays a role in breast cancer development. This sentence should be rephrased (for example: "these processes may play a role in FBC development).

We thank the reviewer for this suggestion. We have rephrased the final sentence in the Abstract to "Both genomic and functional experiments support the concept that cell–cell and cell–extracellular matrix adhesion processes may be disrupted in non-malignant cells of many women who have a family history of breast cancer and that these processes may play a role in FBC development."

2 - Need of a table summarising the information for each cohort (Table S1 is too simplified).

The authors analysed 2 cohorts: Utah (124 women) and Ontario (73 women).

In addition to table S1, there should be a table with better description of all data in each cohort.

The contain the following information:

- How many individuals were grouped into the 6 groups (No FH with and without cancer, BRCA1/2 with and without cancer, BRCA1/2 with and without cancer).*
- For those groups "with cancer", what was the subtype of familial cancer (ER-, HER2-, etc) and what grade?*
- For those groups "with family history" - how many first-degree relatives have been diagnosed with breast cancer? what subtype of breast cancer (ER-, HER2-, etc)?*
- What is the average age per group (ok in figure S1 but should add this info to the table and for both cohorts)*

We have added two supplementary tables (S2 and S3) that indicate how many patients fell into each of the six groups for the Utah and Ontario cohorts, respectively. We also indicate the median age for each group in the Utah cohort. Because these cohorts were recruited retrospectively and due to restrictions in our IRB protocol, we are unable to determine the tumor subtype and grade for these patients. In addition, we are not able to determine the specific number of first-degree relatives who had been diagnosed with breast cancer. However, for patients with a family history of breast cancer, we limited our cohort to those who had at least two first-degree relatives who had been diagnosed with breast cancer. We now state this more explicitly in the Materials and Methods section.

- Which cases were used as controls? Authors say there were 41 from Utah and 13 from Ontario, with no family history of breast cancer. But it is not very clear which group they refer to.

We agree that our prior manuscript was unclear on what the controls were and how we used them. Consequently, we have made substantial revisions to the Materials and Methods section (see subsection entitled "Patient cohorts used for molecular profiling") to clarify these points. We have also updated our schematic diagram (Figure 1A) for clarification.

3 - More concise explanation of the cohorts used in this study.

The authors say " Where possible, patients were matched according to these criteria and by age at which blood was drawn. (...) cohorts included women who either 1) had a family history of breast cancer or did not, 2) carried a pathogenic germline variant in BRCA1 or BRCA2 or did not, and 3) had developed breast cancer or had not"

But no table is provided showing exactly all available information for all individuals considered in this study.

We now provide Tables S2 and S3, which indicate how many samples fell into each of these categories for the Utah and Ontario cohorts.

I find this section of the methods on the study design not very clear. A table summarising clearly what data was used and how the groups were built will help organising this information.

4 - too vague

"Individuals who had not developed breast cancer or who did not have a family history of breast cancer served as controls." (page 6)

How many individuals were controls? Too vague.

As described above, we have extended our explanations of the cohorts and how we used each patient subgroup in the analyses. Tables S2 and S3 provide additional information about the subgroups.

5 - not very explicit

"The samples showed no statistical difference among all groups in covariates, including age." (page 6)

This statement is too vague for a "methods" section. What covariates were tested and how? please explain

6 - more detail needed

"We excluded any gene whose expression correlated with lymphocyte markers or epidemiologic/demographic/health factors"

Not sure what this means and how this analysis was made. What were the epidemiologic/demographic/health factors considered and what do you mean with "correlated"?

Due to the manuscript's size, we previously had included an abbreviated description of these methods in the main manuscript and moved many of the details to Supplementary Methods. We acknowledge that it may not have been clear to the reader that such details were available in Supplementary Methods. The reviewer references the following specific statement: "The samples showed no statistical difference among all groups in covariates, including age." We have simplified this to "The samples showed no statistical difference in age at blood draw." Later in the manuscript, we also now state, "Supplementary Methods contain details about data filtering, normalization, and summarization. Data filtering included a step to identify genes whose expression correlated with lymphocyte markers of epidemiologic/demographic/health factors (Supplementary Methods)." The description in Supplementary Methods now outlines exactly what these lymphocyte markers were, as well as the epidemiologic/demographic/health factors that we evaluated. We also describe the statistical test we used (multifactor ANCOVA test).

7 - reference needed

"We then used the Support Vector Machines (SVM) algorithm to identify multigene patterns that significantly differed between the patient groups." (page 8)

Add citation (was it in R? what package?)

We now provide a citation related to Support Vector Machines (Vapnik, 1998) in the main manuscript and refer the reader to Supplementary Methods, which contains additional references and explicit details about the software tools and settings we used for this part of the analysis.

For all data analysis in R, please provide Rmarkdown or knitR files to reproduce your results.

We agree with the reviewer on the importance of reproducibility. To make it easier for others to reproduce our analysis, we have created an IPython notebook that includes shell commands, R code, and Python scripts that we used to filter key pathways and to create the analysis figures we present in the paper. We preferred to use IPython rather than knitr because it was easier to combine these various types of code and scripts. We have attached this notebook as well as the underlying data and script files as a supplementary file in the paper (replacing the previous Supplementary Data File 1, which contained a subset of what we now provide).

8 - link leads to too many undocumented data files

"All software scripts and code that were used to execute these analyses can be accessed at <https://github.com/srp33/BCSP>" (page 9)

In this link there are 9 folders, one of them is called "scripts". The "scripts" folder contains 21 files of (I guess) scripts. The names of the files are "preparefastq", "prepend_string_to_file", "decompress", "delete_temp_files", etc. There is another subfolder in "scripts", with an extra 20 data files.

The idea of making the research reproducible by sharing the code is a great one. The authors could also include some documentation to explain the user what folder/file(s) they should run to reproduce a each section of the paper.

Previously, we included a link to a repository that included code for multiple research projects. We have created a new repository that is specific to this manuscript and have updated the link to this repository in the manuscript. In addition, we have added a README file that provides an overview of 1) prerequisite software that must be installed to execute the analysis, 2) notes about accessing the raw data, and 3) scripts that a reader could use to execute the analysis.

9 - Some figures need to explicitly say what was used as a "control":

- In figure 3 and 5 what are the "controls"?

(For example, for figure 3, are "controls" women who did not had a family history of breast cancer?)

Add this information to both Figure 3 and Figure 5 caption to be more explicit.

To make this clearer, we have updated Figures 3-5 and S6 so that they more explicitly describe the populations that were compared. We have also updated the captions. We no longer use the term "Control" in these figures or in the captions. Instead we describe the samples as coming from individuals who did or did not have a family history of breast cancer, etc. We are confident that these changes will make these figures easier to interpret.

10 - Discussion

Instead of "Together our findings suggest that aberrant signaling within cancer pathways in peripheral blood may be a sign of eventual tumor development in general." (page 17)

A bit too vague. Instead of "a sign of eventual tumor development in general", the authors should say more clearly that disruption of cell adhesion pathways is associated to higher risk of developing familial breast cancer.

We thank the reviewer for this suggestion. We agree, and we have modified the Discussion accordingly.

2nd Editorial Decision

01 October 2015

Thank you again for submitting your work to Molecular Systems Biology. First of all, I would like to apologize for the delayed response, which was due to the late arrival of one of the referee reports. We sent the manuscript to the same reviewers who evaluated the earlier version of your work (MSB-14-5798) and we have now heard back from them. As you will see below, the reviewers appreciate that the study has been improved. However, Reviewer #2 still raises a number of significant concerns, which should be carefully addressed in a revision of the manuscript.

In particular, Reviewer #2 raised the following issues:

- i) s/he remains unconvinced that the multiple comparisons problem has been solved
- ii) s/he mentions that a formal distinction between the training and testing sets needs to be made in all cases
- iii) s/he thinks that all analyses need to be repeated by separating four groups (namely BRCA1/2 No cancer, BRCA1/2 No cancer, BRCA1/2 cancer, BRCA1/2 cancer -as in Figure 2-).

These issues are rather fundamental and therefore they need to be convincingly addressed.

We have circulated the reports to both reviewers as part of our 'pre-decision cross-commenting' policy. During this process, Reviewer #1 made the following constructive comments, which are pasted below for your reference:

" - Is the sample size of both replication cohorts (Lim / Bellacosa) sufficient to validate the effect size of the association signal? If not, then a combined P-value should be devised based on both replication cohorts (similar to meta analysis of independent association studies and/or two-stage GWAS studies, i.e. discovery and replication). Overall, every replication cohort should in theory validate the primary association signal under the same P-value cutoff. However, if the cohort size is

too small to validate the primary association signal, then I would advise to combine the two datasets (Lim and Bellacosa) into one combined dataset and perform a meta-analysis for replication (still at the same P-value cutoff).

Also, I agree with Reviewer #2 that multiple-testing correction should be applied to claim a given pathway to be significantly replicated. However, Bonferroni correction might be indeed too stringent since genes are shared across the 45 pathways. In the latter case I would correct P-values using the FDR procedure and estimate the proportion of true positives (1- π_0 statistics) using Storey's q-value approach.

- Repeating the analyses by separating four groups: Genetic risk for familial breast cancer is polygenic, yet the individual risk factors might channel on the same downstream pathways. In addition, it is not clear whether all BRCAX patients are actually negative for BRCA mutations, making a clear-cut separation actually difficult. Perhaps some BRCAX patients have pathogenic cis regulatory variants that mimic the effects of a BRCA1/2 coding mutation and have been missed. Yet I agree that it would be advisable to, at least, assess how many differentially regulated pathways distinguish the following patient groups:

- 1) BRCAX no cancer vs BRCA1/2 no cancer and
- 2) BRCAX cancer vs BRCA1/2 cancer

If the authors hypothesis remains true (that cell adhesion pathway genes are dysregulated in women with cancer), then they should not separate patients in either comparisons (1) or (2).

- Regarding Figure 2: it would be indeed appropriate to plot all data separately for the training and testing cohorts (i.e. Utah and Ontario separately)."

REFeree COMMENTS

Reviewer #1:

- The authors have addressed all my comments.

- In response to comment #2 (Rev #1). This analysis, and the two (out of 45) significant pathways with differential coding mutation rates between BRCAX and BRCA1/2 patients, should be added to the manuscript. Please indicate the direction (higher/lower mutation rates in BRCAX patients) and effect sizes. This information may actually be relevant for readers that would like to map out the causal genetic factors leading to perturbation of these pathways in FBC-BRCAX patients. The relatively weak enrichment of coding germline variants among the remaining 43 studied pathways in FBC-BRCAX patients indicates that other genetic mechanisms may act on these pathways (e.g. regulatory variants; risk variants in genes that are indirectly connected to the studied pathways).

- The P-value shown in Figure 5A ($p=0.02$) does not match with the P-value in the Results section ($p=0.04$). Please clarify.

- Figure 2: "for which" appears twice in the figure legend.

Reviewer #2:

The major concern in this manuscript identified by all three reviewers is whether the multiple comparisons problem has been dealt with in an adequate manner. The authors argue in their response to reviewers somewhat unconvincingly that the sets have been kept independent and the predictors locked prior to the first time that the "test" sets were assessed. For example, the authors state that leave one out cross validation was used in the Bellacosa and Lim sets. The predictors should have been locked prior to testing these sets.

Further, a correction factor is needed in the test sets given the fairly large number of profiles (45 different profiles) that were tested. While it is acceptable to argue that Bonferroni is too stringent, it

is not appropriate to argue that the multiple comparisons problem was dealt with by using multiple sets.

While the authors attempt to define the goal for the project, they miss two key points of clinical relevance. First women with aberrations in known familial genes are at increased risk. The question in this group is whether this risk is/will be manifest or not. The question in the group without these known genomic aberrations is whether there are additional familial risk factors that can be assessed by the pathway analysis. Lumping the two groups in the majority of the studies does not approach the key and important questions in the field. We can already identify many high risk families by testing for known risk factor genes.

Thus for each major figure, the analysis should be separated as it is in figure 2. BRCA1/2 No cancer, BRCA1/2 cancer and the comparisons done for all of the combinations with statistical significance.

In figure 2, were the Utah and Ontario cohorts mixed. The figure legend indicates that both are included but there is only one dataset presented. These should be kept separate if there is a rigorous test and training approach used.

Reviewer #3:

Overall the authors did a good job in addressing the comments we made in the initial round of review. I have three minor comments remaining:

The authors correct and address all comments made about the fact that some sentences were too vague. The writing is overall more accurate throughout the manuscript. Table 1 and Table 2 really help in understanding the cohorts. The text is also explicit now

The authors did address the comment we made about the fact that they should be more explicit when they mention 'controls' in Figure 3; but again in Figure 5 and in the figure caption the authors refer again to 'controls' but it is not clear what the control is.

Figure 5B. What does the Y-axis represent? The caption says figure 5B shows cells responsiveness to PF573228 but I am not sure what the values represent in the Y-axis. The X-axis says "Normal" and "Prophylactic" but the text mentions FBC women versus no-FBC...

1st Revision - authors' response

30 December 2015

Dr. Polychronidou's comments: *Thank you again for submitting your work to Molecular Systems Biology. First of all, I would like to apologize for the delayed response, which was due to the late arrival of one of the referee reports. We sent the manuscript to the same reviewers who evaluated the earlier version of your work (MSB-14-5798) and we have now heard back from them. As you will see below, the reviewers appreciate that the study has been improved. However, Reviewer #2 still raises a number of significant concerns, which should be carefully addressed in a revision of the manuscript.*

In particular, Reviewer #2 raised the following issues:

- i) s/he remains unconvinced that the multiple comparisons problem has been solved*
 - ii) s/he mentions that a formal distinction between the training and testing sets needs to be made in all cases*
 - iii) s/he thinks that all analyses need to be repeated by separating four groups (namely BRCA1/2 no cancer, BRCA1/2 cancer, BRCA1/2 No cancer, BRCA1/2 cancer -as in Figure 2-).*
- These issues are rather fundamental and therefore they need to be convincingly addressed.*

We have circulated the reports to both reviewers as part of our 'pre-decision cross-commenting'

policy. During this process, Reviewer #1 made the following constructive comments, which are pasted below for your reference:

" - Is the sample size of both replication cohorts (Lim / Bellacosa) sufficient to validate the effect size of the association signal? If not, then a combined P-value should be devised based on both replication cohorts (similar to meta analysis of independent association studies and/or two-stage GWAS studies, i.e. discovery and replication). Overall, every replication cohort should in theory validate the primary association signal under the same P-value cutoff. However, if the cohort size is too small to validate the primary association signal, then I would advise to combine the two datasets (Lim and Bellacosa) into one combined dataset and perform a meta-analysis for replication (still at the same P-value cutoff).

We thank the editor for summarizing these points. We have addressed these comments in the responses below.

First, we would like to clarify that the goal of our genomic analyses was to generate robust hypotheses about biological processes that may influence familial breast cancer development. We focused on pathways that were consistently and statistically significant across all 5 genomic data sets that we analyzed, after multiple rounds of stringent filtering. These results formed the basis for our cell-based, laboratory tests, which support the hypothesis that cell-adhesion processes influence familial breast cancer development.

We appreciate the recommendation by the reviewer and editor to use a ‘meta analysis’ approach as an alternative to our stepwise, rank-based method. As recommended, we combined p-values across data sets using three such methods. (Note: we used this approach rather than the alternative of combining the Lim and Bellacosa datasets directly; the latter approach is challenging as they use different microarray platforms.) We corrected the meta p-values for multiple testing using Storey’s q-value method. We found that this approach resulted in a larger number of candidate pathways than the rank-based approach that we used originally (Langfelder & Horvath, 2008). Our approach likely yields fewer pathways because it requires significance consistently across all datasets, whereas the meta-analysis approach is less conservative and may indicate overall significance, even when significance has been obtained only for a subset of the datasets. Furthermore, the same pathways that we had identified using the rank-based approach were identified as significant using the meta-analysis approach and were among the top pathways identified with this approach, thus validating the conservative stability and robustness of our rank-based approach. In the manuscript, we now provide results both for the rank-based method and for the most conservative of the meta-analysis methods (described below). Code that was used for this analysis is provided in *Computer Code EV2*.

Because a variety of meta-analysis techniques have been developed over the years to combine p-values across heterogeneous data sets, we employed three different methods to ensure that our findings were not particular to one technique. These techniques included R. A. Fisher’s combined probability test, which uses a “sum of logs” approach to merge p-values. We also applied Wilkinson’s method, which uses the Beta distribution to infer the probability of obtaining a given number of significant results by chance. And we used the simpler “mean p” method. In all cases, we used the implementation of these methods provided in the *metap* statistical package (<https://cran.r-project.org/web/packages/metap/metap.pdf>). In addition, for each of these methods, we corrected for multiple tests using Storey’s q-value method. We found that even after stringent filtering (p-value < 0.05 and q-value < 0.05), each of these methods identified a larger number of significant pathways than the rank-based p-value approach (Langfelder & Horvath, 2008) that we used originally. More importantly, we found that the top pathways identified with all of these approaches were similar (see Datasets EV1,5). This finding coincides with the intent of our study, which was to use our genomic analyses as a way to generate and filter hypotheses about which pathways may play a role in familial breast cancer risk. Our focus was to identify consistently significant candidates for lab-based validation rather than to make strong assertions based on our genomic findings alone.

Reviewer #2 asked that we make a more formal distinction between the training and testing sets that we used. We are now more careful to indicate how pathway-based predictions were generated

and evaluated for each data set (for example, see updated Figure 1).

We have also addressed Reviewer #2's request that we repeat the analyses by separating the patient data into four groups: BRCAX no cancer, BRCA1/2 no cancer, BRCAX cancer, and BRCA1/2 cancer. We describe these results (comparing no cancer vs. cancer samples) in the manuscript and have included them as Datasets EV 6-7. Some of the pathways that we had identified in our original analyses (when BRCA1/2 and BRCAX are combined) attained statistical significance for either BRCA1/2 or BRCAX when compared separately. This suggests that different biological processes may influence familial breast cancer risk, depending on whether an individual carries a BRCA1/2 mutation. However, for both groups (BRCA1/2 and BRCAX), a variety of cell-adhesion-related pathways, including those we highlight in the manuscript, attained statistical significance. Samples sizes were relatively small for these subgroup comparisons; thus it is unknown to what level decreased sample sizes may have led to greater variation in these results.

Also, I agree with Reviewer #2 that multiple-testing correction should be applied to claim a given pathway to be significantly replicated. However, Bonferroni correction might be indeed too stringent since genes are shared across the 45 pathways. In the latter case I would correct P-values using the FDR procedure and estimate the proportion of true positives (1-pi0 statistics) using Storey's q-value approach.

As described above, we used meta-analysis approaches to calculate p-values that represent combined evidence across multiple data sets, and we applied Storey's q-value approach to correct these results for multiple comparisons. We again emphasize 1) that these approaches resulted in a larger number of candidate pathways than the rank-based approach that we used originally (Langfelder & Horvath, 2008) and 2) that the same pathways that we had identified using the rankbased approach were identified as significant using the meta-analysis approach and were among the top pathways identified with this alternative approach. However, to better justify our approach and to appropriately respond to the editor/reviewer critiques, we have included both sets of results with our manuscript.

- Repeating the analyses by separating four groups: Genetic risk for familial breast cancer is polygenic, yet the individual risk factors might channel on the same downstream pathways. In addition, it is not clear whether all BRCAX patients are actually negative for BRCA mutations, making a clear-cut separation actually difficult. Perhaps some BRCAX patients have pathogenic cis regulatory variants that mimic the effects of a BRCA1/2 coding mutation and have been missed. Yet I agree that it would be advisable to, at least, assess how many differentially regulated pathways distinguish the following patient groups:

- 1) BRCAX no cancer vs BRCA1/2 no cancer and
- 2) BRCAX cancer vs BRCA1/2 cancer

If the authors hypothesis remains true (that cell adhesion pathway genes are dysregulated in women with cancer), then they should not separate patients in either comparisons (1) or (2).

The BRCAX patients are indeed negative for BRCA1/2 mutations, according to commercial genetic testing described in the manuscript. However, as the editor points out, it is possible that some BRCAX patients have pathogenic *cis* regulatory variants that mimic the effects of BRCA1/2 coding mutations. Based on the editor's comments, we have also searched for pathways that showed significant differences between *BRCAX no cancer* and *BRCA1/2 no cancer* individuals or between *BRCAX cancer* and *BRCA1/2 cancer* individuals. When working with the mutation data, we excluded pathways that showed statistical significance in either direction. We have also provided supplementary data files (see Datasets EV8-9) that indicate which genes carried mutations in these pathways for these groups. These filtering steps have modestly reduced the number of significant pathways in our final results. However, our conclusions remain unchanged.

- Regarding Figure 2: it would be indeed appropriate to plot all data separately for the training and testing cohorts (i.e. Utah and Ontario separately)."

Thank you for this suggestion. We have modified Figure 2 so that it shows expression levels for the Utah and Ontario data sets separately. We also now indicate the number of samples in each group.

Reviewer #1:

- *The authors have addressed all my comments.*

We thank the reviewer for taking time to review and comment carefully on our manuscript.

- *In response to comment #2 (Rev #1). This analysis, and the two (out of 45) significant pathways with differential coding mutation rates between BRCA1/2 and BRCA1/2 patients, should be added to the manuscript. Please indicate the direction (higher/lower mutation rates in BRCA1/2 patients) and effect sizes. This information may actually be relevant for readers that would like to map out the causal genetic factors leading to perturbation of these pathways in FBC-BRCA1/2 patients. The relatively weak enrichment of coding germline variants among the remaining 43 studied pathways in FBC-BRCA1/2 patients indicates that other genetic mechanisms may act on these pathways (e.g. regulatory variants; risk variants in genes that are indirectly connected to the studied pathways).*

We thank the reviewer for these suggestions. We have searched for pathways that showed significant differences between *BRCA1/2 no cancer* and *BRCA1/2 cancer* individuals or between *BRCA1/2 no cancer* and *BRCA1/2 cancer* individuals. When working with the mutation data, we excluded pathways that showed statistical significance in either direction. We have also provided supplementary data files that indicate which genes carried mutations in these pathways for these groups. Effect sizes can be inferred from these data. These filtering steps have modestly reduced the number of significant pathways in our final results. However, our conclusions remain unchanged; our final pathway list still points consistently toward cell adhesion processes, which we validated using lab-based tests.

- *The P-value shown in Figure 5A ($p=0.02$) does not match with the P-value in the Results section ($p=0.04$). Please clarify.*

Thank you for this comment. The p-value shown in Figure 5A ($p=0.02$) was correct. We have updated this in the manuscript.

- *Figure 2: "for which" appears twice in the figure legend.*

Thank you for noticing this. We have corrected it in the manuscript.

Reviewer #2:

The major concern in this manuscript identified by all three reviewers is whether the multiple comparisons problem has been dealt with in an adequate manner. The authors argue in their response to reviewers somewhat unconvincingly that the sets have been kept independent and the predictors locked prior to the first time that the "test" sets were assessed. For example, the authors state that leave one out cross validation was used in the Bellacosa and Lim sets. The predictors should have been locked prior to testing these sets.

We thank the reviewer for taking time to comment so carefully on our manuscript. When analyzing the Utah and Ontario gene-expression data, which characterized expression levels in peripheral blood mononuclear cells, we performed cross validation within the Utah data and then derived a model from the Utah data and made predictions for the Ontario samples. This helped to ensure that the signals we identified in the Utah data were generalizable to patients from a different geographical region and helped us to avoid overfitting. We used leave-one-out cross validation for the Lim and Bellacosa data sets because they represent different cell types (mammary epithelial cells) and were profiled on different microarray platforms than the peripheral blood mononuclear cells. The broader goal of these genomic analyses was to generate robust hypotheses about biological processes that may influence familial breast cancer development. We focused on pathways that were consistently and statistically significant across all 5 genomic data sets, after multiple rounds of stringent filtering. These results formed the basis for our cell-based, laboratory tests, which support the hypothesis that cell-adhesion processes influence familial breast cancer development. We now address these points more thoroughly in the manuscript. In particular, we

have updated Figure 1, our flow diagram, to illustrate that the purpose of our genomic analyses was to generate hypotheses for our lab-based validations, rather than to stand on their own.

Further, a correction factor is needed in the test sets given the fairly large number of profiles (45 different profiles) that were tested. While it is acceptable to argue that Bonferoni is too stringent, it is not appropriate to argue that the multiple comparisons problem was dealt with by using multiple sets.

We thank the reviewer for this comment. As described above in our comments to the editor, we combined p-values across data sets using a “meta analysis” approach and compared them against the results of the rank-based approach. These findings support our original conclusions.

While the authors attempt to define the goal for the project, they miss two key points of clinical relevance. First women with aberrations in known familial genes are at increased risk. The question in this group is whether this risk is/will be manifest or not. The question in the group without these known genomic aberrations is whether there are additional familial risk factors that can be assessed by the pathway analysis. Lumping the two groups in the majority of the studies does not approach the key and important questions in the field. We can already identify many high risk families by testing for known risk factor genes. Thus for each major figure, the analysis should be separated as it is in figure 2. BRCAX no cancer, BRCA1/2 No cancer, BRCAX cancer BRCA1/2 cancer and the comparisons done for all of the combinations with statistical significance.

The original intent of our study was to identify those signaling events that influence familial breast cancer risk, independent of BRCA mutation status. Importantly, although BRCA1/2 mutation status does provide clinically relevant information on its own, a considerable proportion of women who carry these mutations never develop breast cancer. Indeed, our study provides insight into biological processes that may influence familial breast cancer development, in addition to, or in complement to, DNA repair processes. However, in response to this question from the reviewer, we have repeated the analyses by separating the Utah/Ontario patient data into four groups: BRCAX no cancer, BRCA1/2 no cancer, BRCAX cancer, and BRCA1/2 cancer. We have described these results in our response to the editor (see above).

In figure 2, were the Utah and Ontario cohorts mixed. The figure legend indicates that both are included but there is only one dataset presented. These should be kept separate if there is a rigorous test and training approach used.

Thank you for this suggestion. We have modified Figure 2 so that it shows expression levels for the Utah and Ontario data sets separately. We also now indicate the number of samples in each group.

Reviewer #3:

Overall the authors did a good job in addressing the comments we made in the initial round of review.

We thank the reviewer for taking time to review and comment carefully on our manuscript.

I have three minor comments remaining:

The authors correct and address all comments made about the fact that some sentences were too vague. The writing is overall more accurate throughout the manuscript. Table 1 and Table 2 really help in understanding the cohorts. The text is also explicit now.

We thank the reviewer for these positive comments.

The authors did address the comment we made about the fact that they should be more explicit when they mention 'controls' in Figure 3; but again in Figure 5 and in the figure caption the authors refer again to 'controls' but it is not clear what the control is.

We thank the reviewer for this feedback. We have updated the title and caption for Figure 5 so that it is much more descriptive and no longer uses the term *control*.

Figure 5B. What does the Y-axis represent? The caption says figure 5B shows cells responsiveness to PF573228 but I am not sure what the values represent in the Y-axis. The X-axis says "Normal" and "Prophylactic" but the text mentions FBC women versus no-FBC...

Thank you for this suggestion. We have updated the figure caption to more accurately reflect that we were comparing normal breast-epithelial cells from women who had undergone prophylactic surgery against cells from women who had undergone breast-reduction surgery for reasons unrelated to breast cancer risk. We also indicate that the response values on the y-axis indicate the drug concentration that induces a response that reaches half of its maximal effect. This metric is commonly used to measure a drug's efficacy.

3rd Editorial Decision

08 February 2016

Thank you again for submitting your work to Molecular Systems Biology. We have now heard back from the two referees who were asked to evaluate your manuscript. As you will see below, the referees are now satisfied with the modifications made. However, referee #1 mentions a remaining issue, which we would ask you to address in a revision.

 REFEREE COMMENTS

Reviewer #1:

I would like to thank the authors for performing the additional experiments (i.e. summary statistics and for providing additional data for the FBC-BRCAX patients). They have strengthened their conclusions, yet their remains an important issue with regard to the meta P-value analysis that must be corrected.

The authors have evaluated the robustness of their rank-based pathway enrichment results by combining P-values across datasets with the "mean P value" approach (see Results section and Dataset EV5). This meta analysis method requires at least four P-values to compute a meta P-value, yet their Dataset EV5 contains only three input sets (i.e. Utah RNA p-value, Ontario RNA p-value, and Utah DNA p-value). It is therefore unclear how the authors have computed meta P-values with the "meanp" approach.

The source code, which the authors have kindly provided (this is much appreciated!) in "Computer Code EV2", has the actual function (CalculateMetaP.R, see below) and command line (Novel_signaling_pathways_underlying_familial_breast_cancer_susceptibility.ipynb, see below) to calculate meta P-values. Both files suggest that the meta P-values were calculated with another meta P value method, namely "sump", which only requires three input P-values (i.e. Utah RNA p-value, Ontario RNA p-value, and Utah DNA p-value). Manual recalculation for the top three pathways confirms that the used meta P-value method was "sump" and not "meanp". Neither the Results nor the Methods section describe that this method was used for combining P-values.

The other described methods were the classical Fisher's method (implemented as sumlog in the metap R package) and the method proposed by Wilkinson (implemented as wilkinsonp in the metap R package), yet the results of these two methods are not presented in the manuscript.

```
# From CalculateMetaP.R
calcMetaP = function(x)
{
y = x[which(!is.na(x))]
```

```

sump(y)$p # 131
# sumlog(y)$p #314
# wilkinsonp(y)$p #307
## sumz(y)$p
## votep(y)$p
}

# From Novel_signaling_pathways_underlying_familial_breast_cancer_susceptibility.ipynb
Rscript --vanilla code/CalculateMetaP.R Results/${comparison}_Summary.txt 3,4,5
Results/UtahOntario_${comparison}_Meta.txt

```

In conclusion, I am generally in favor of publishing this MS, after the aforementioned important error has been fixed.

Reviewer #2:

The authors have greatly improved the manuscript in response to the reviews. The authors have emphasized in their response to reviewers that the manuscript should be viewed as "hypothesis generating". In this manner, the manuscript has provided a number of interesting hypotheses that warrant further investigation. It is important to note that differences between the approaches used and the structures of the data sets preclude a formal training and test set approach. Thus the data should be considered as exploratory. As such the data and concepts should be made available to the research community for further exploration. As such I recommend publication.

2nd Revision - authors' response

09 February 2016

Reviewer #1

I would like to thank the authors for performing the additional experiments (i.e. summary statistics and for providing additional data for the FBC-BRCAX patients). They have strengthened their conclusions, yet their remains an important issue with regard to the meta P-value analysis that must be corrected.

We thank the reviewer for a careful review of our manuscript and of the accompanying code. Below we describe how we have addressed this concern.

The authors have evaluated the robustness of their rank-based pathway enrichment results by combining P-values across datasets with the "mean P value" approach (see Results section and Dataset EV5). This meta analysis method requires at least four P-values to compute a meta P-value, yet their Dataset EV5 contains only three input sets (i.e. Utah RNA p-value, Ontario RNA p-value, and Utah DNA p-value). It is therefore unclear how the authors have computed meta P-values with the "meanp" approach.

The source code, which the authors have kindly provided (this is much appreciated!) in "Computer Code EV2", has the actual function (CalculateMetaP.R, see below) and command line (Novel_signaling_pathways_underlying_familial_breast_cancer_susceptibility.ipynb, see below) to calculate meta P-values. Both files suggest that the meta P-values were calculated with another meta P value method, namely "sump", which only requires three input P-values (i.e. Utah RNA p-value, Ontario RNA p-value, and Utah DNA p-value). Manual recalculation for the top three pathways confirms that the used meta P-value method was "sump" and not "meanp". Neither the Results nor the Methods section describe that this method was used for combining P-values.

The reviewer is correct. We did indeed use the “sump” method rather than the “meanp” method to calculate the meta P-values. Originally, we had considered using the “meanp” method before we realized that it did not meet the criteria of at least four input P-values. In revising the manuscript, we confused these two methods. We have fixed this description in the manuscript. We thank the reviewer for making us aware of this discrepancy!

The other described methods were the classical Fisher's method (implemented as sumlog in the metap R package) and the method proposed by Wilkinson (implemented as wilkinsonp in the metap R package), yet the results of these two methods are not presented in the manuscript.

From CalculateMetaP.R

```
calcMetaP = function(x)
```

```
{
```

```
y = x[which(!is.na(x))]
```

```
sump(y)$p # 131
```

```
# sumlog(y)$p #314
```

```
# wilkinsonp(y)$p #307
```

```
## sumz(y)$p
```

```
## votep(y)$p
```

```
}
```

```
# From Novel_signaling_pathways_underlying_familial_breast_cancer_susceptibility.ipynb
```

```
Rscript --vanilla code/CalculateMetaP.R Results/${comparison}_Summary.txt 3,4,5
```

```
Results/UtahOntario_${comparison}_Meta.txt
```

We have modified the appendix so that it now includes results for all three meta P-value methods that we applied (Fisher’s combined probability test, Wilkinson’s method, and the “sum p” approach). We have also modified the wording in the manuscript to reflect this change. The updated version of our Jupyter notebook (see Computer Code EV2) has also been updated. Below we indicate the locations in the manuscript that were changed to accommodate this change.

- Page 17, lines 2-3
- Page 24, lines 12-17.
- Page 24, line 21: Datasets EV6-7 -> Datasets EV8-9
- Page 25, line 4: Datasets EV8-9 -> Datasets EV10-11
- Page 25, line 10: Dataset EV10 -> Dataset EV12
- Page 25, line 17: Datasets EV11-14 -> Datasets EV13-16

In conclusion, I am generally in favor of publishing this MS, after the aforementioned important error has been fixed.

We thank the reviewer for these positive comments.

Reviewer #2:

The authors have greatly improved the manuscript in response to the reviews. The authors have emphasized in their response to reviewers that the manuscript should be viewed as "hypothesis generating". In this manner, the manuscript has provided a number of interesting hypotheses that warrant further investigation. It is important to note that differences between the approaches used and the structures of the data sets preclude a formal training and test set approach. Thus the data should be considered as exploratory. As such the data and concepts should be made available to the research community for further exploration. As such I recommend publication.

We thank the reviewer for providing a careful review of our manuscript! We acknowledge that our findings require further exploration. To this end, we have made our data and code available to the research community.